# A Sharp Error Analysis for the Fused Lasso, with Application to Approximate Changepoint Screening

**Kevin Lin**
Carnegie Mellon University
Pittsburgh, PA 15213
kevinl1@andrew.cmu.edu

**James Sharpnack**
University of California, Davis
Davis, CA 95616
jsharpna@ucdavis.edu

**Alessandro Rinaldo**
Carnegie Mellon University
Pittsburgh, PA 15213
arinaldo@stat.cmu.edu

**Ryan J. Tibshirani**
Carnegie Mellon University
Pittsburgh, PA 15213
ryantibs@stat.cmu.edu

## Abstract

In the 1-dimensional multiple changepoint detection problem, we derive a new fast error rate for the fused lasso estimator, under the assumption that the mean vector has a sparse number of changepoints. This rate is seen to be suboptimal (compared to the minimax rate) by only a factor of $\log \log n$. Our proof technique is centered around a novel construction that we call a *lower interpolant*. We extend our results to misspecified models and exponential family distributions. We also describe the implications of our error analysis for the approximate screening of changepoints.

## 1 Introduction

Consider the 1-dimensional multiple changepoint model

$$y_i = \theta_{0,i} + \epsilon_i, \quad i = 1, \ldots, n, \tag{1}$$

where $\epsilon_i$, $i = 1, \ldots, n$ are i.i.d. errors, and $\theta_{0,i}$, $i = 1, \ldots, n$ is a piecewise constant mean sequence, having a set of changepoints

$$S_0 = \left\{ i \in \{1, \ldots, n-1\} : \theta_{0,i} \neq \theta_{0,i+1} \right\}. \tag{2}$$

This is a well-studied setting, and there is a large body of literature on estimation of the piecewise constant mean vector $\theta_0 \in \mathbb{R}^n$ and its changepoints $S_0$ using various estimators; refer, e.g., to the surveys Brodsky and Darkhovski (1993); Chen and Gupta (2000); Eckley et al. (2011).

In this work, we consider the *1-dimensional fused lasso* (also called 1d fused lasso, or simply fused lasso) estimator, which, given a data vector $y \in \mathbb{R}^n$ from a model as in (1), is defined by

$$\widehat{\theta} = \operatorname*{argmin}_{\theta \in \mathbb{R}^n} \frac{1}{2} \sum_{i=1}^{n} (y_i - \theta_i)^2 + \lambda \sum_{i=1}^{n-1} |\theta_i - \theta_{i+1}|, \tag{3}$$

where $\lambda \geq 0$ serves as a tuning parameter. This was proposed and named by Tibshirani et al. (2005), but the same idea was proposed earlier in signal processing, under the name *total variation denoising*, by Rudin et al. (1992). Variants of the fused lasso have been used in biology to detect regions where two genomic samples differ due to genetic variations (Tibshirani and Wang, 2008), in finance to detect shifts in the stock market (Chan et al., 2014), and in neuroscience to detect changes in stationary behaviors of the brain (Aston and Kirch, 2012). Popularity of the fused lasso can be attributed in part to its computational scalability, the optimization problem in (3) being convex and highly structured. There has also been plenty of supporting statistical theory developed for the fused lasso, which we review in Section 2.

**Notation.** We will make use of the following quantities that are defined in terms of the mean $\theta_0$ in (1) and its changepoint set $S_0$ in (2). We denote the size of the changepoint set by $s_0 = |S_0|$. We enumerate $S_0 = \{t_1, \ldots, t_{s_0}\}$, where $1 \leq t_1 < \ldots < t_{s_0} < n$, and for convenience we set $t_0 = 0$, $t_{s_0+1} = n$. The smallest distance between changepoints in $\theta_0$ is denoted by

$$W_n = \min_{i=0,1\ldots,s_0} (t_{i+1} - t_i), \tag{4}$$

and the smallest distance between consecutive levels of $\theta_0$ by

$$H_n = \min_{i \in S_0} |\theta_{0,i+1} - \theta_{0,i}|. \tag{5}$$

We use $D \in \mathbb{R}^{(n-1)\times n}$ to denote the difference operator

$$D = \begin{bmatrix} -1 & 1 & 0 & \ldots & 0 \\ 0 & -1 & 1 & \ldots & 0 \\ \vdots & & \ddots & \ddots & \\ 0 & 0 & \ldots & -1 & 1 \end{bmatrix}. \tag{6}$$

Note that $s_0 = \|D\theta_0\|_0$. We write $D_S$ to extract rows of $D$ indexed by a subset $S \subseteq \{1, \ldots, n-1\}$, and $D_{-S}$ to extract the rows in $S^c = \{1, \ldots, n-1\} \setminus S$.

For a vector $x \in \mathbb{R}^n$, we use $\|x\|_n^2 = \|x\|_2^2/n$ to denote its length-scaled $\ell_2$ norm. For sequences $a_n, b_n$, we use standard asymptotic notation: $a_n = O(b_n)$ to denote that $a_n/b_n$ is bounded for large enough $n$, $a_n = \Omega(b_n)$ to denote that $b_n/a_n$ is bounded for large enough $n$, $a_n = \Theta(b_n)$ to denote that both $a_n = O(b_n)$ and $a_n = \Omega(b_n)$, $a_n = o(b_n)$ to denote that $a_n/b_n \to 0$, and $a_n = \omega(b_n)$ to denote that $b_n/a_n \to 0$. For random sequences $A_n, B_n$, we write $A_n = O_{\mathbb{P}}(B_n)$ to denote that $A_n/B_n$ is bounded in probability. A random variable $Z$ is said to have a sub-Gaussian distribution provided that $\mathbb{E}(Z) = 0$ and $\mathbb{P}(|Z| > t) \leq 2\exp(-t^2/(2\sigma^2))$ for all $t \geq 0$, and a constant $\sigma > 0$.

**Summary of results.** Our main focus is on deriving a sharp estimation error bound for the fused lasso, parametrized by the number of changepoints $s_0$ in $\theta_0$. We also study several consequences of our error bound and its analysis. A summary of our contributions is as follows.

- **New error analysis for the fused lasso.** In Section 3, we develop a new error analysis for the fused lasso, in the model (1) with sub-Gaussian errors. Our analysis leverages a novel quantity that we call a *lower interpolant* to approximate the fused lasso estimate (once it has been orthogonalized with respect to the changepoint structure of the mean $\theta_0$) with $2s_0 + 2$ monotonic segments, which allows for finer control of the empirical process term.

  When $s_0 = O(1)$, and the changepoint locations in $S_0$ are (asymptotically) evenly spaced, our main result implies $\mathbb{E}\|\widehat{\theta} - \theta_0\|_n^2 = O(\log n(\log\log n)/n)$ for the fused lasso estimator $\widehat{\theta}$ in (3). This is slower than the minimax rate by a $\log\log n$ factor. Our result improves on previously established results from Dalalyan et al. (2017), and after the completion of this paper, was itself improved upon by Guntuboyina et al. (2017) (who are able to remove the extraneous $\log\log n$ factor).

- **Extension to misspecified and exponential family models.** In Section 4, we extend our error analysis to cover a mean vector $\theta_0$ that is not necessarily piecewise constant (or in other words, has potentially many changepoints). In Section 5, we extend our analysis to exponential family models. The latter extension, especially, is of practical importance, as many applications, e.g., CNV data analysis, call for changepoint detection on count data.

- **Application to approximate screening and recovery.** In Section 6, we establish that the maximum distance between any true changepoint and its nearest estimated changepoint is $O_{\mathbb{P}}(\log n(\log\log n)/H_n^2)$ using the fused lasso, when $s_0 = O(1)$ and all changepoints are (asymptotically) evenly spaced. After applying simple post-processing step, we show that the maximum distance between any estimated changepoint and its nearest true changepoint is of the same order. Our proof technique relies only on the estimation error rate of the fused lasso, and therefore immediately generalizes to *any* estimator of $\theta_0$, where the distance (for approximate changepoint screening and recovery) is a function of the inherent error rate.

The supplementary document gives numerical simulations that support the theory in this paper.

## 2 Preliminary review of existing theory

We begin by describing known results on the quantity $\|\widehat{\theta} - \theta_0\|_n^2$, the estimation error between the fused lasso estimate $\widehat{\theta}$ in (3) and the mean $\theta_0$ in (1).

Early results on the fused lasso are found in Mammen and van de Geer (1997) (see also Tibshirani (2014) for a translation to a setting more consistent with that of the current paper). These authors study what may be called the *weak sparsity* case, in which it is that assumed $\|D\theta_0\|_1 \leq C_n$, with $D$ being the difference operator in (6). Assuming additionally that the errors in (1) are sub-Gaussian, Mammen and van de Geer (1997) show that for a choice of tuning parameter $\lambda = \Theta(n^{1/3}C_n^{-1/3})$, the fused lasso estimate $\widehat{\theta}$ in (3) satisfies

$$\|\widehat{\theta} - \theta_0\|_n^2 = O_{\mathbb{P}}(n^{-2/3}C_n^{2/3}). \tag{7}$$

The weak sparsity setting is not the focus of our paper, but we still recall the above result to give a sense of the difference between the weak and *strong sparsity* settings, the latter being the setting in which we assume control over $s_0 = \|D\theta_0\|_0$, as we do in the current paper. Prior to this paper, the strongest result in the strong sparsity setting was given by Dalalyan et al. (2017), who assume $N(0, \sigma^2)$ errors in (1), and show that for $\lambda = \sigma\sqrt{2n\log(n/\delta)}$, the fused lasso estimate satisfies

$$\|\widehat{\theta} - \theta_0\|_n^2 \leq C\sigma^2 \frac{s_0 \log(n/\delta)}{n}\left(\log n + \frac{n}{W_n}\right), \tag{8}$$

with probability at least $1 - 2\delta$, for large enough $n$, and a constant $C > 0$, where recall $W_n$ is the minimum distance between changepoints in $\theta_0$, as in (4). Our main result in Theorem 1 improves upon (8) in two ways: by reducing the first $\log n$ term inside the brackets to $\log s_0 + \log\log n$, and reducing the second $n/W_n$ term to $\sqrt{n/W_n}$.

After our paper was completed, Guntuboyina et al. (2017) gave an even sharper error rate for the fused lasso (and more broadly, for trend the family of higher-order filtering estimates as defined in Steidl et al. (2006); Kim et al. (2009); Tibshirani (2014)). Again assuming $N(0, \sigma^2)$ errors in (1), as well as $W_n \geq cn/(s_0 + 1)$ for some constant $c \geq 1$, these authors show that the family of fused lasso estimates $\{\widehat{\theta}_\lambda, \lambda \geq 0\}$ (using subscripts here to explicitly denote the dependence on the tuning parameter $\lambda$) satisfies

$$\inf_{\lambda \geq 0} \|\widehat{\theta}_\lambda - \theta_0\|_n^2 \leq C\sigma^2 \frac{s_0 + 1}{n}\log\left(\frac{en}{s_0 + 1}\right) + \frac{4\sigma^2\delta}{n}, \tag{9}$$

with probability at least $1 - \exp(-\delta)$, for large enough $n$, and a constant $C > 0$. The above bound is sharper than ours in Theorem 1 in that $(\log s_0 + \log\log n)\log n + \sqrt{n/W_n}$ is replaced essentially by $\log W_n$. (Also, the result in (9) does not actually require $W_n \geq cn/(s_0 + 1)$, but only requires the distance between changepoints where jumps alternate in sign to be larger than $cn/(s_0 + 1)$, which is another improvement.) Further comparisons will be made in Remark 1 following Theorem 1.

There are numerous other estimators, e.g., based on segmentation techniques or wavelets, that admit estimation results comparable to those above. These are described in Remark 2 following Theorem 1. Lastly, it can be seen the minimax estimation error over the class of signals $\theta_0$ with $s_0$ changepoints, assuming $N(0, \sigma^2)$ errors in (1), satisfies

$$\inf_{\widehat{\theta}} \sup_{\|D\theta_0\|_0 \leq s_0} \mathbb{E}\|\widehat{\theta} - \theta_0\|_n^2 \geq C\sigma^2 \frac{s_0}{n}\log\left(\frac{n}{s_0}\right), \tag{10}$$

for large enough $n$, and a constant $C > 0$. This says that one cannot hope to improve the rate in (9). The minimax result in (10) follows from standard minimax theory for sparse normal means problems, as in, e.g., Johnstone (2015); for a proof, see Padilla et al. (2016).

## 3 Sharp error analysis for the fused lasso estimator

Here we derive a sharper error bound for the fused lasso, improving upon the previously established result of Dalalyan et al. (2017) as stated in (8). Our proof is based on a concept that we call a *lower interpolant*, which as far as we can tell, is a new idea that may be of interest in its own right.

**Theorem 1.** *Assume the data model in* (1), *with errors $\epsilon_i$, $i = 1, \ldots, n$ i.i.d. from a sub-Gaussian distribution. Then under a choice of tuning parameter $\lambda = (nW_n)^{1/4}$, the fused lasso estimate $\widehat{\theta}$ in* (3) *satisfies*

$$\|\widehat{\theta} - \theta_0\|_n^2 \leq \gamma^2 c \frac{s_0}{n} \left( (\log s_0 + \log \log n) \log n + \sqrt{\frac{n}{W_n}} \right),$$

*with probability at least $1 - \exp(-C\gamma)$, for all $\gamma > 1$ and $n \geq N$, where $c, C, N > 0$ are constants that depend on only $\sigma$ (the parameter appearing in the sub-Gaussian distribution of the errors).*

An immediate corollary is as follows.

**Corollary 1.** *Under the same assumptions as in Theorem 1, we have*

$$\mathbb{E}\|\widehat{\theta} - \theta_0\|_n^2 \leq c \frac{s_0}{n} \left( (\log s_0 + \log \log n) \log n + \sqrt{\frac{n}{W_n}} \right),$$

*for some constant $c > 0$.*

We give some remarks comparing Theorem 1 to related results in the literature.

**Remark 1** (**Comparison to Dalalyan et al. (2017); Guntuboyina et al. (2017)**). *We can see that the result in Theorem 1 is sharper than that in* (8) *from Dalalyan et al. (2017) for any $s_0, W_n$, as $\log s_0 \leq \log n$ and $\sqrt{n/W_n} \leq n/W_n$. Moreover, when $s_0 = O(1)$ and $W_n = \Theta(n)$, the rates are $\log^2 n/n$ and $\log n(\log \log n)/n$ from Theorem 1 and* (8), *respectively.*

*Comparing the result in Theorem 1 to that in* (9) *from Guntuboyina et al. (2017), the latter is sharper in that it reduces the factor of $(\log s_0 + \log \log n) \log n + \sqrt{n/W_n}$ to a single term of $\log W_n$. In the case $s_0 = O(1)$ and $W_n = \Theta(n)$, the rates are $\log n(\log \log n)/n$ and $\log n/n$ from Theorem 1 and* (8), *respectively, and the latter rate cannot be improved, owing to the minimax lower bound in* (10). *Similar to our expectation bound in Corollary 1, Guntuboyina et al. (2017) establish*

$$\inf_{\lambda \geq 0} \mathbb{E}\|\widehat{\theta}_\lambda - \theta_0\|_n^2 \leq C\sigma^2 \frac{s_0 + 1}{n} \log\left(\frac{en}{s_0 + 1}\right), \tag{11}$$

*for the family of fused lasso estimates $\{\widehat{\theta}_\lambda, \lambda \geq 0\}$, for large enough $n$, and a constant $C > 0$. Like their high probability result in* (9), *their expectation result in* (11) *is stated in terms of an infimum over $\lambda \geq 0$, and does not provide an explicit value of $\lambda$ that attains the bound. (Inspection of their proofs suggests that it is not at all easy to make such a value of $\lambda$ explicit.) Meanwhile, Theorem 1 and Corollary 1 have the advantage this choice is made explicit, as in $\lambda = (nW_n)^{1/4}$.*

**Remark 2** (**Comparison to other estimators**). *Various other estimators obtain comparable estimation error rates. In what follows, all results are stated in the case $s_0 = O(1)$. The Potts estimator, defined by replacing the $\ell_1$ penalty $\sum_{i=1}^{n-1} |\theta_i - \theta_{i+1}|$ in* (3) *with the $\ell_0$ penalty $\sum_{i=1}^{n-1} 1\{\theta_i \neq \theta_{i+1}\}$, and denoted say by $\widehat{\theta}^{\mathrm{Potts}}$, satisfies a bound $\|\widehat{\theta}^{\mathrm{Potts}} - \theta_0\|_n^2 = O(\log n/n)$ a.s. as shown by Boysen et al. (2009). Wavelet denoising (placing weak conditions on the wavelet basis), denoted by $\widehat{\theta}^{\mathrm{wav}}$, satisfies $\mathbb{E}\|\widehat{\theta}^{\mathrm{wav}} - \theta_0\|_n^2 = O(\log^2 n/n)$ as shown by Donoho and Johnstone (1994). Pairing unbalanced Haar (UH) wavelets with a basis selection method, Fryzlewicz (2007) developed an estimator $\widehat{\theta}^{\mathrm{UH}}$ with $\mathbb{E}\|\widehat{\theta}^{\mathrm{UH}} - \theta_0\|_n^2 = O(\log^2 n/n)$. Though they are not written in this form, the results in Fryzlewicz (2016) imply that his "tail-greedy" unbalanced Haar (TGUH) estimator, $\widehat{\theta}^{\mathrm{TGUH}}$, satisfies $\|\widehat{\theta}^{\mathrm{TGUH}} - \theta_0\|_n^2 = O(\log^2 n/n)$ with probability tending to 1.*

Here is an overview of the proof of Theorem 1. The full proof is deferred until the supplement, as with all proofs in this paper. We begin by deriving a basic inequality (stemming from the optimality of the fused lasso estimate $\widehat{\theta}$ in (3)):

$$\|\widehat{\theta} - \theta_0\|_2^2 \leq 2\epsilon^\top(\widehat{\theta} - \theta_0) + 2\lambda\left(\|D\theta_0\|_1 - \|D\widehat{\theta}\|_1\right). \tag{12}$$

To precisely control the empirical process term $\epsilon^\top(\widehat{\theta} - \theta_0)$, we consider a decomposition

$$\epsilon^\top(\widehat{\theta} - \theta_0) = \epsilon^\top \widehat{\delta} + \epsilon^\top \widehat{x},$$

where we define $\widehat{\delta} = P_0(\widehat{\theta} - \theta_0)$ and $\widehat{x} = P_1\widehat{\theta}$. Here $P_0$ is the projection matrix onto the piecewise constant structure inherent in $\theta_0$, and $P_1 = I - P_0$. More precisely, writing $S_0 = \{t_1, \ldots, t_{s_0}\}$ for the set of ordered changepoints in $\theta_0$, we define $B_j = \{t_j + 1, \ldots, t_{j+1}\}$, and denote by $\mathbb{1}_{B_j} \in \mathbb{R}^n$

the indicator of block $B_j$, for $j = 0, \ldots, s_0$. In this notation, $P_0$ is the projection onto the $(s_0 + 1)$-dimensional linear subspace $\mathcal{R} = \mathrm{span}\{\mathbb{1}_{B_0}, \ldots, \mathbb{1}_{B_{s_0}}\}$. The parameter $\widehat{\delta}$ lies in an low-dimensional subspace, which makes bounding the term $\epsilon^\top \widehat{\delta}$ relatively easy. Bounding the term $\epsilon^\top \widehat{x}$ requires a much more intricate argument, which is spelled out in the following lemmas.

Lemma 1 is a deterministic result ensuring the existence of what we call a *lower interpolant* $\widehat{z}$ to $\widehat{x}$. This interpolant approximates $\widehat{x}$ using $2s_0 + 2$ monotonic segments, and its empirical process term $\epsilon^\top \widehat{z}$ can be finely controlled, as shown in Lemma 2. The residual from the interpolant approximation, denoted $\widehat{w} = \widehat{x} - \widehat{z}$, has an empirical process term $\epsilon^\top \widehat{w}$ that is more crudely controlled, in Lemma 3. Put together, as in $\epsilon^\top \widehat{x} = \epsilon^\top \widehat{z} + \epsilon^\top \widehat{w}$, gives the final control on $\epsilon^\top \widehat{x}$.

Before stating Lemma 1, we define the class of vectors containing the lower interpolant. Given any collection of changepoints $t_1 < \ldots < t_{s_0}$ (and $t_0 = 0$, $t_{s_0+1} = n$), let $\mathcal{M}$ be the set of "piecewise monotonic" vectors $z \in \mathbb{R}^n$, with the following properties, for each $i = 0, \ldots, s_0$:

(i) there exists a point $t_i'$ such that $t_i + 1 \leq t_i' \leq t_{i+1}$, and such that the absolute value $|z_j|$ is nonincreasing over the segment $j \in \{t_i + 1, \ldots, t_i'\}$, and nondecreasing over the segment $j \in \{t_i', \ldots, t_{i+1}\}$;

(ii) the signs remain constant on the monotone pieces,

$$\mathrm{sign}(z_{t_i}) \cdot \mathrm{sign}(z_j) \geq 0, \quad j = t_i + 1, \ldots, t_i',$$
$$\mathrm{sign}(z_{t_{i+1}}) \cdot \mathrm{sign}(z_j) \geq 0, \quad j = t_i' + 1, \ldots, t_{i+1}.$$

Now we state our lemma that characterizes the lower interpolant.

**Lemma 1.** *Given changepoints $t_0 < \ldots < t_{s_0+1}$, and any $x \in \mathbb{R}^n$, there exists a vector $z \in \mathcal{M}$ (not necessarily unique), such that the following statements hold:*

$$\|D_{-S_0} x\|_1 = \|D_{-S_0} z\|_1 + \|D_{-S_0}(x - z)\|_1, \tag{13}$$

$$\|D_{S_0} x\|_1 = \|D_{S_0} z\|_1 \leq \|D_{-S_0} z\|_1 + \frac{4\sqrt{s_0}}{\sqrt{W_n}} \|z\|_2, \tag{14}$$

$$\|z\|_2 \leq \|x\|_2 \quad and \quad \|x - z\|_2 \leq \|x\|_2, \tag{15}$$

*where $D \in \mathbb{R}^{(n-1) \times n}$ is the difference matrix in (6). We call a vector $z$ with these properties a* lower interpolant *to $x$.*

Loosely speaking, the lower interpolant $\widehat{z}$ can be visualized by taking a string that lies initially on top of $\widehat{x}$, is nailed down at the changepoints $t_0, \ldots t_{s_0+1}$, and then pulled taut while maintaining that it is not greater (elementwise) than $\widehat{x}$, in magnitude. Here "pulling taut" means that $\|D\widehat{z}\|_1$ is made small. Figure 1 provides illustrations of the interpolant $\widehat{z}$ to $\widehat{x}$ for a few examples.

Note that $\widehat{z}$ consists of $2s_0 + 2$ monotonic pieces. This special structure leads to a sharp concentration inequality. The next lemma is the primary contributor to the fast rate given in Theorem 1.

**Lemma 2.** *Given changepoints $t_1 < \ldots < t_{s_0}$, there exists constants $c_I, C_I, N_I > 0$ such that when $\epsilon \in \mathbb{R}^n$ has i.i.d. sub-Gaussian components,*

$$\mathbb{P}\left( \sup_{z \in \mathcal{M}} \frac{|\epsilon^\top z|}{\|z\|_2} > \gamma c_I \sqrt{(\log s_0 + \log\log n) s_0 \log n} \right) \leq 2\exp\left( -C_I \gamma^2 c_I^2 (\log s_0 + \log\log n) \right),$$

*for any $\gamma > 1$, and $n \geq N_I$.*

Finally, the following lemma controls the residuals, $\widehat{w} = \widehat{x} - \widehat{z}$.

**Lemma 3.** *Given changepoints $t_1 < \ldots < t_{s_0}$, there exists constants $c_R, C_R > 0$ such that when $\epsilon \in \mathbb{R}^n$ has i.i.d. sub-Gaussian components,*

$$\mathbb{P}\left( \sup_{w \in \mathcal{R}^\perp} \frac{|\epsilon^\top w|}{\sqrt{\|D_{-S_0} w\|_1 \|w\|_2}} > \gamma c_R (n s_0)^{1/4} \right) \leq 2\exp(-C_R \gamma^2 c_R^2 \sqrt{s_0}),$$

*for any $\gamma > 1$, where $\mathcal{R}^\perp$ is the orthogonal complement of $\mathcal{R} = \mathrm{span}\{\mathbb{1}_{B_0}, \ldots, \mathbb{1}_{B_{s_0}}\}$.*

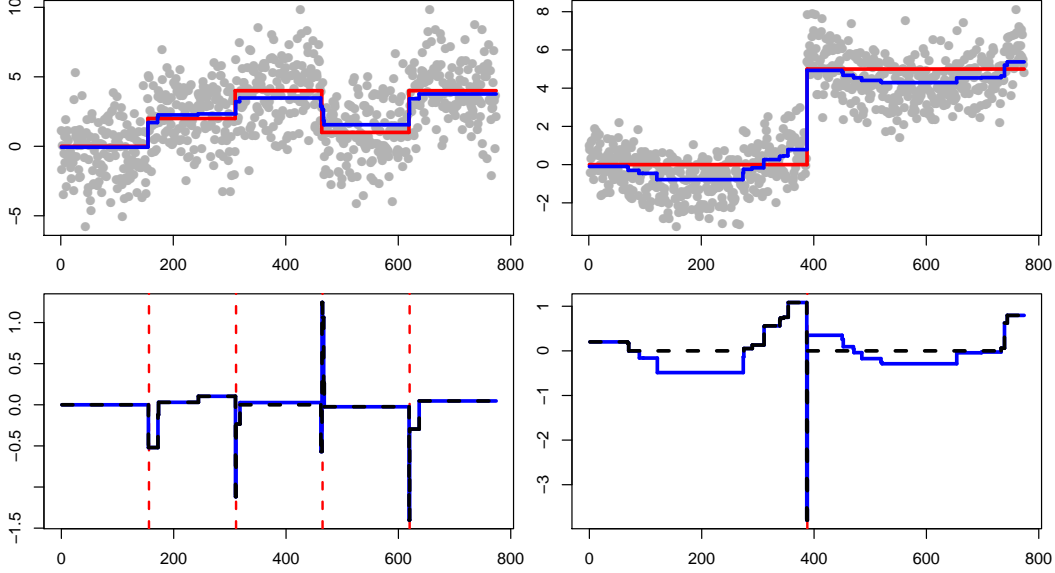

Figure 1: *The lower interpolants for two examples (in the left and right columns), each with $n = 800$ points. In the top row, the data $y$ (in gray) and underlying signal $\theta_0$ (red) are plotted across the locations $1, \ldots, n$. Also shown is the fused lasso estimate $\widehat{\theta}$ (blue). In the bottom row, the error vector $\widehat{x} = P_1 \widehat{\theta}$ is plotted (blue) as well as the interpolant (black), and the dotted vertical lines (red) denote the changepoints $t_1, \ldots t_{s_0}$ of $\theta_0$.*

## 4 Extension to misspecified models

We consider data from the model in (1) but where the mean $\theta_0$ is not necessarily piecewise constant (i.e., where $s_0$ is potentially large). Let us define

$$\theta_0(s) = \operatorname*{argmin}_{\theta \in \mathbb{R}^n} \|\theta_0 - \theta\|_2^2 \quad \text{subject to} \quad \|D\theta\|_0 \leq s, \tag{16}$$

which we call the *best $s$-approximation* to $\theta_0$. We now present an extension of Theorem 1.

**Theorem 2.** *Assume the data model in (1), with errors $\epsilon_i$, $i = 1, \ldots, n$ i.i.d. from a sub-Gaussian distribution. For any $s$, consider the best $s$-approximation $\theta_0(s)$ to $\theta_0$, as in (16), and let $W_n(s)$ be the minimum distance between the $s$ changepoints in $\theta_0(s)$. Then under a choice of tuning parameter $\lambda = (nW_n(s))^{1/4}$, the fused lasso estimate $\widehat{\theta}$ in (3) satisfies*

$$\|\widehat{\theta} - \theta_0\|_n^2 \leq \|\theta_0(s) - \theta_0\|_n^2 + \gamma^2 c \frac{s}{n} \left( (\log s + \log \log n) \log n + \sqrt{\frac{n}{W_n(s)}} \right), \tag{17}$$

*with probability at least $1 - \exp(-C\gamma)$, for all $\gamma > 1$ and $n \geq N$, where $c, C, N > 0$ are constants that depend on only $\sigma$. Further, if $\lambda$ is chosen large enough so that $\|D\widehat{\theta}\|_0 \leq s$ on an event $E$, then*

$$\|\widehat{\theta} - \theta_0(s)\|_n^2 \leq \gamma^2 c \frac{s}{n} \left( (\log s + \log \log n) \log n + \frac{\lambda^2}{W_n(s)} + \frac{n}{\lambda^2} \right), \tag{18}$$

*on $E$ intersected with an event of probability at least $1 - \exp(-C\gamma)$, for all $\gamma > 1$, $n \geq N$, where $c, C, N > 0$ are the same constants as above.*

The first result in (17) in Theorem 2 is a standard oracle inequality. It provides a bound on the error of the fused lasso estimator that decomposes into two parts, the first term being the approximation error, determined by the proximity of $\theta_0(s)$ to $\theta_0$, and second term being the usual bound we would encounter if the mean truly had $s$ changepoints.

The second result in (18) in the theorem is a direct bound on the estimation error $\|\widehat{\theta} - \theta_0(s)\|_n^2$. We see that the estimation error can be small, apparently regardless of the size of $\|\theta_0(s) - \theta_0\|_n^2$, if we take $\lambda$ to be large enough for $\widehat{\theta}$ to itself have $s$ changepoints. But the rate worsens as $\lambda$ grows larger, so implicitly, the proximity of $\theta_0(s)$ to $\theta_0$ does play an role (if $\theta_0$ were actually far away from a signal with $s$ changepoints, then we may have to take $\lambda$ very large to ensure that $\widehat{\theta}$ has $s$ changepoints).

**Remark 3** (**Comparison to other results**). *Dalalyan et al. (2017); Guntuboyina et al. (2017) also provide oracle inequalities and their results could be adapted to take forms as in Theorem 2. It is not clear to us that previous results on other estimators, such as those from Remark 2, adapt as easily.*

## 5    Extension to exponential family models

We consider data $y = (y_1, \ldots, y_n) \in \mathbb{R}^n$ with independent components distributed according to

$$p(y_i; \theta_{0,i}) = h(y_i) \exp\big(y_i \theta_{0,i} - \Lambda(\theta_{0,i})\big), \quad i = 1, \ldots, n. \tag{19}$$

Here, for each $i = 1, \ldots, n$, the parameter $\theta_{0,i}$ is the natural parameter in the exponential family and $\Lambda$ is the cumulant generating function. As before, in the location model, we are mainly interested in the case in which the natural parameter vector $\theta_0$ is piecewise constant (with $s_0$ denoting its number of changepoints, as before). Estimation is now based on penalization of the negative log-likelihood:

$$\widehat{\theta} = \operatorname*{argmin}_{\theta \in \mathbb{R}^n} \sum_{i=1}^{n} \big( -y_i \theta_i + \Lambda(\theta_i) \big) + \lambda \sum_{i=1}^{n} |\theta_i - \theta_{i+1}|, \tag{20}$$

Since the cumulant generating function $\Lambda$ is always convex in exponential families, the above is a convex optimization problem. We present an estimation error bound the present setting.

**Theorem 3.** *Assume the data model in* (19)*, with a strictly convex, twice continuously differentiable cumulant generating function $\Lambda$. Assume that $\theta_{0,i} \in [l, u]$, $i = 1, \ldots, n$ for constants $l, u \in \mathbb{R}$, and add the constraints $\theta_i \in [l, u]$, $i = 1, \ldots, n$ in the optimization problem in* (20)*. Finally, assume that the random variables $y_i - \mathbb{E}(y_i)$, $i = 1, \ldots, n$ obey a sub-Gaussian distribution, with parameter $\sigma$. Then under a choice of tuning parameter $\lambda = (nW_n)^{1/4}$, the exponential family fused lasso estimate $\widehat{\theta}$ in* (20) *(subject to the additional boundedness constraints) satisfies*

$$\|\widehat{\theta} - \theta_0\|_n^2 \le \gamma^2 c \frac{s_0}{n} \left( (\log s_0 + \log \log n) \log n + \sqrt{\frac{n}{W_n}} \right),$$

*with probability at least $1 - \exp(-C\gamma)$, for all $\gamma > 1$ and $n \ge N$, where $c, C, N > 0$ are constants that depend on only $l, u, \sigma$.*

**Remark 4** (**Roles of $l, u$**). *The restriction of $\theta_{0,i}$ and the optimization parameters in* (20) *to $[l, u]$, for $i = 1, \ldots, n$, is used to ensure that the second derivative of $\Lambda$ is bounded away from zero. (The same property could be accomplished by instead adding a small squared $\ell_2$ penalty on $\theta$ in* (20)*.) A more refined analysis could alleviate the need for this bounded domain (or extra squared $\ell_2$ penalty) but we do not pursue this for simplicity.*

**Remark 5** (**Sub-Gaussianity in exponential families**). *When are the random variables $y_i - \mathbb{E}(y_i)$, $i = 1, \ldots, n$ sub-Gaussian, in an exponential family model* (19)*? A simple sufficient condition (not specific to exponential families, in fact) is that these centered variates are bounded. This covers the binomial model $y_i \sim \mathrm{Bin}(k, \mu(\theta_{0,i}))$, where $\mu(\theta_{0,i}) = 1/(1 + e^{-\theta_{0,i}})$, $i = 1, \ldots, n$, and $k$ is a fixed constant. Hence Theorem 3 applies to binomial data.*

*For Poisson data $y_i \sim \mathrm{Pois}(\mu(\theta_{0,i}))$, where $\mu(\theta_{0,i}) = e^{\theta_{0,i}}$, $i = 1, \ldots, n$, we now give two options for the analysis. The first is to assume a maximum achieveable count (which may be reasonable in CNV data) and then apply Theorem 3 owing again to boundedness. The second is to invoke the fact that Poisson random variables have sub-exponential (rather than sub-Gaussian) tails, and then use a truncation argument, to show that for the Poisson fused lasso estimate $\widehat{\theta}$ in* (20) *(under the additional boundedness constraints), with $\lambda = \log n (nW_n)^{1/4}$,*

$$\|\widehat{\theta} - \theta_0\|_n^2 \le \gamma^2 c \frac{s_0 \log n}{n} \left( (\log s_0 + \log \log n) \log n + \sqrt{\frac{n}{W_n}} \right), \tag{21}$$

*with probability at least $1 - \exp(-C\gamma) - 1/n$, for all $\gamma > 1$ and $n \ge N$, where $c, C, N > 0$ are constants depending on $l, u$. This is slower than the rate in Theorem 3 by a factor of $\log n$.*

**Remark 6** (**Comparison to other results**). *The results in Dalalyan et al. (2017); Guntuboyina et al. (2017) assume normal errors. It seems believable to us that the results of Dalalyan et al. (2017) could be extended to sub-Gaussian errors and hence exponential family data, in a manner similar to what we have done above in Theorem 3. To us, this is less clear for the results of Guntuboyina et al. (2017), which rely on some technical calculations involving Gaussian widths. It is even less clear to us how results from other estimators, as in Remark 2, extend to exponential family data.*

# 6 Approximate changepoint screening and recovery

In many applications of changepoint detection, one may be interested in estimation of the changepoint locations in $\theta_0$, rather than the mean vector $\theta_0$ as a whole. In this section, we show that estimation of the changepoint locations and of $\theta_0$ itself are two very closely linked problems, in the following sense: any procedure with guarantees on its error in estimating $\theta_0$ automatically has certain approximate changepoint detection guarantees, and not surprisingly, a faster error rate (in estimating $\theta_0$) translates into a stronger statement about approximate changepoint detection. We use this general link to prove new approximate changepoint screening results for the fused lasso. We also show that in general a simple post-processing step may be used to discard spurious detected changepoints, and again apply this to the fused lasso to yield new approximate changepoint recovery results.

It helps to introduce some additional notation. For a vector $\theta \in \mathbb{R}^n$, we write $S(\theta)$ for the set of its changepoint indices, i.e.,

$$S(\theta) = \big\{ i \in \{1, \ldots, n-1\} : \theta_i \neq \theta_{i+1} \big\}.$$

Recall, we abbreviate $S_0 = S(\theta_0)$ for the changepoints of the underlying mean $\theta_0$. For two discrete sets $A, B$, we define the metrics

$$d(A|B) = \max_{b \in B} \min_{a \in A} |a - b| \quad \text{and} \quad d_H(A, B) = \max\big\{ d(A|B), d(B|A) \big\}.$$

The first metric above can be seen as a one-sided screening distance from $B$ to $A$, measuring the furthest distance of an element in $B$ to its closest element in $A$. The second metric above is known as the *Hausdorff distance* between $A$ and $B$.

**Approximate changepoint screening.**   We present our general theorem on changepoint screening. The basic idea behind the result is quite simple: if an estimator misses a (large) changepoint in $\theta_0$, then its estimation error must suffer, and we can use this fact to bound the screening distance.

**Theorem 4.** *Let $\widetilde{\theta} \in \mathbb{R}^n$ be an estimator such that $\|\widetilde{\theta} - \theta_0\|_n^2 = O_{\mathbb{P}}(R_n)$. Assume that $nR_n/H_n^2 = o(W_n)$, where, recall, $H_n$ is the minimum gap between adjacent levels of $\theta_0$, defined in (5), and $W_n$ is the minimum distance between adjacent changepoints of $\theta_0$, defined in (4). Then*

$$d\big(S(\widetilde{\theta}) \,|\, S_0\big) = O_{\mathbb{P}}\bigg( \frac{nR_n}{H_n^2} \bigg).$$

**Remark 7** (**Generic setting: no specific data model, and no assumptions on estimator**). *Importantly, Theorem 4 assumes no data model whatsoever, and treats $\widetilde{\theta}$ as a generic estimator of $\theta_0$. (Of course, through the statement $\|\widetilde{\theta} - \theta_0\|_n^2 = O_{\mathbb{P}}(R_n)$, one can see that $\widetilde{\theta}$ is random, constructed from data that depends on $\theta_0$, but no specific data model is required, nor are any specific properties of $\widetilde{\theta}$, other than its error rate.) This flexibility allows for the result to be applied in any problem setting in which one has control of the error in estimating a piecewise constant parameter $\theta_0$ (in some cases this may be easier to obtain, compared to direct analysis of detection properties). A similar idea was used (concurrently and independently) by Fryzlewicz (2016) in the analysis of the TGUH estimator.*

Combining the above theorem with known error rates for the fused lasso estimator—(7) in the weak sparsity case, and Theorem 1 in the strong sparsity case—gives the following result.

**Corollary 2.** *Assume the data model in (1), with errors $\epsilon_i$, $i = 1, \ldots, n$ i.i.d. from a sub-Gaussian distribution. Let $C_n = \|D\theta_0\|_1$, and assume that $H_n = \omega(n^{1/6} C_n^{1/3}/\sqrt{W_n})$. Then the fused lasso estimator $\widehat{\theta}$ in (3) with $\lambda = \Theta(n^{1/3} C_n^{-1/3})$ satisfies*

$$d\big(S(\widehat{\theta}) \,|\, S_0\big) = O_{\mathbb{P}}\bigg( \frac{n^{1/3} C_n^{2/3}}{H_n^2} \bigg). \tag{22}$$

*Alternatively, assume $s_0 = O(1)$, $W_n = \Theta(n)$, and $H_n = \omega(\sqrt{\log n(\log \log n)/n})$. Then the fused lasso with $\lambda = \Theta(\sqrt{n})$ satisfies*

$$d\big(S(\widehat{\theta}) \,|\, S_0\big) = O_{\mathbb{P}}\bigg( \frac{\log n(\log \log n)}{H_n^2} \bigg). \tag{23}$$

**Remark 8** (**Changepoint detection limit**). *The restriction $H_n = \omega(\sqrt{\log n(\log \log n)/n})$ for (23) in Corollary 2 is very close to the optimal detection limit of $H_n = \omega(1/\sqrt{n})$: Duembgen and Walther (2008) showed that in Gaussian changepoint model with a single elevated region, and $W_n = \Theta(n)$, there is no test for detecting a changepoint that has asymptotic power 1 unless $H_n = \omega(1/\sqrt{n})$.*

Combining Theorem 4 with (21) gives the following (a similar result holds for the binomial model).

**Corollary 3.** *Assume $y_i \sim \mathrm{Pois}(e^{\theta_{0,i}})$, independently, for $i = 1, \ldots, n$, and assume $\|\theta_0\|_\infty = O(1)$, $s_0 = O(1)$, $W_n = \Theta(n)$, $H_n = \omega(\log n \sqrt{\log \log n/n})$. Then for the Poisson fused lasso estimator $\widehat{\theta}$ in (20) (subject to appropriate boundedness constraints) with $\lambda = \Theta(\log n \sqrt{n})$, we have*

$$d\big(S(\widehat{\theta}) \,|\, S_0\big) = O_{\mathbb{P}}\Big(\frac{\log^2 n(\log \log n)}{H_n^2}\Big).$$

**Approximate changepoint recovery.** We present a post-processing procedure for the estimated changepoints in $\widetilde{\theta}$, to eliminate changepoints of $\widetilde{\theta}$ that lie far away from changepoints of $\theta_0$. Our procedure is based on convolving $\widetilde{\theta}$ with a filter that resembles the mother Haar wavelet. Consider

$$F_i(\widetilde{\theta}) = \frac{1}{b_n} \sum_{j=i+1}^{i+b_n} \widetilde{\theta}_j - \frac{1}{b_n} \sum_{j=i-b_n+1}^{i} \widetilde{\theta}_j, \quad \text{for } i = b_n, \ldots, n - b_n, \tag{24}$$

for an integral bandwidth $b_n > 0$. By evaluating the filter $F_i(\widetilde{\theta})$ at all locations $i = b_n, \ldots, n - b_n$, and retaining only locations at which the filter value is large (in magnitude), we can approximately recovery the changepoints of $\theta_0$, in the Hausdorff metric.

**Theorem 5.** *Let $\widetilde{\theta} \in \mathbb{R}^n$ be such that $\|\widetilde{\theta} - \theta_0\|_n^2 = O_{\mathbb{P}}(R_n)$. Consider the following procedure: we evaluate the filter in (24) with bandwidth $b_n$ at locations in*

$$I_F(\widetilde{\theta}) = \Big\{ i \in \{b_n, \ldots, n - b_n\} : i \in S(\widetilde{\theta}), \text{ or } i + b_n \in S(\widetilde{\theta}), \text{ or } i - b_n \in S(\widetilde{\theta}) \Big\} \cup \{b_n, n - b_n\},$$

*and define a set of filtered points $S_F(\widetilde{\theta}) = \{i \in I_F(\widetilde{\theta}) : |F_i(\widetilde{\theta})| \geq \tau_n\}$, for a threshold level $\tau_n$. If $b_n, \tau_n$ satisfy $b_n = \omega(nR_n/H_n^2)$, $2b_n \leq W_n$, and $\tau_n/H_n \to \rho \in (0, 1)$ as $n \to \infty$, then*

$$\mathbb{P}\Big(d_H\big(S_F(\widetilde{\theta}), S_0\big) \leq 2b_n\Big) \to 1 \quad \text{as } n \to \infty.$$

Note that the set of filtered points $|S_F(\widetilde{\theta})|$ in Theorem 5 is not necessarily of a subset of the original set of estimated changepoints $S(\widetilde{\theta})$, but it has the property $|S_F(\widetilde{\theta})| \leq 3|S(\widetilde{\theta})| + 2$.

We finish with corollaries for the fused lasso. For space reasons, remarks comparing them to related approximate recovery results in the literature are deferred to the supplement.

**Corollary 4.** *Assume the data model in (1), with errors $\epsilon_i$, $i = 1, \ldots, n$ i.i.d. from a sub-Gaussian distribution. Let $C_n = \|D\theta_0\|_1$. If we apply the post-processing procedure in Theorem 5 to the fused lasso estimator $\widehat{\theta}$ in (3) with $\lambda = \Theta(n^{1/3}C_n^{-1/3})$, $b_n = \lfloor n^{1/3}C_n^{2/3}\nu_n^2/H_n^2 \rfloor \leq W_n/2$ for a sequence $\nu_n \to \infty$, and $\tau_n/H_n \to \rho \in (0, 1)$, then*

$$\mathbb{P}\Big(d_H\big(S_F(\widehat{\theta}), S_0\big) \leq \frac{2n^{1/3}C_n^{2/3}\nu_n^2}{H_n^2}\Big) \to 1 \quad \text{as } n \to \infty. \tag{25}$$

*Alternatively, assuming $s_0 = O(1)$, $W_n = \Theta(n)$, if we apply the same post-processing procedure to the fused lasso with $\lambda = \Theta(\sqrt{n})$, $b_n = \lfloor \log n(\log \log n)\nu_n^2/H_n^2 \rfloor \leq W_n/2$ for a sequence $\nu_n \to \infty$, and $\tau_n/H_n \to \rho \in (0, 1)$, then*

$$\mathbb{P}\Big(d_H\big(S_F(\widehat{\theta}), S_0\big) \leq \frac{2\log n(\log \log n)\nu_n^2}{H_n^2}\Big) \to 1 \quad \text{as } n \to \infty. \tag{26}$$

**Corollary 5.** *Assume $y_i \sim \mathrm{Pois}(e^{\theta_{0,i}})$, independently, for $i = 1, \ldots, n$, and assume $\|\theta_0\|_\infty = O(1)$, $s_0 = O(1)$, $W_n = \Theta(n)$. If we apply the post-processing method in Theorem 5 to the Poisson fused lasso estimator $\widehat{\theta}$ in (20) (subject to appropriate boundedness constraints) with $\lambda = \Theta(\log n \sqrt{n})$, $b_n = \lfloor \log^2 n(\log \log n)\nu_n^2/H_n^2 \rfloor \leq W_n/2$ for a sequence $\nu_n \to \infty$, and $\tau_n/H_n \to \rho \in (0, 1)$, then*

$$\mathbb{P}\Big(d_H\big(S_F(\widehat{\theta}), S_0\big) \leq \frac{2\log^2 n(\log \log n)\nu_n^2}{H_n^2}\Big) \to 1 \quad \text{as } n \to \infty.$$

## 7 Summary

We gave a new error analysis for the fused lasso, with extensions to misspecified models and data from exponential families. We showed that error bounds for general changepoint estimators lead to approximate changepoint screening results, and after post-processing, approximate recovery results.

**Acknolwedgements.** JS was supported by NSF Grant DMS-1712996. RT was supported by NSF Grant DMS-1554123.

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
