[Supplementary Material]

# Supplement to "A Sharp Error Analysis for the Fused Lasso, with Application to Approximate Changepoint Screening"

**Kevin Lin**
Carnegie Mellon University
Pittsburgh, PA 15213
kevinl1@andrew.cmu.edu

**James Sharpnack**
University of California, Davis
Davis, CA 95616
jsharpna@ucdavis.edu

**Alessandro Rinaldo**
Carnegie Mellon University
Pittsburgh, PA 15213
arinaldo@stat.cmu.edu

**Ryan J. Tibshirani**
Carnegie Mellon University
Pittsburgh, PA 15213
ryantibs@stat.cmu.edu

We give proofs for the results in "A Sharp Error Analysis for the Fused Lasso, with Application to Approximate Changepoint Screening". We also provide numerical simulations that support some of our theoretical results.

## A.1 Proofs of Theorem 1 and Corollary 1

We denote by $N(r, S, \|\cdot\|)$ the covering number of a set $S$ in a norm $\|\cdot\|$, i.e., the smallest number of $\|\cdot\|$-balls of radius $r$ needed to cover $S$. We call $\log N(r, S, \|\cdot\|)$ the log covering or entropy number. Recall that we write $\|\cdot\|_n = \|\cdot\|_2/\sqrt{n}$ for the scaled $\ell_2$ norm, and that we say a random variable $Z$ has a sub-Gaussian distribution provided that

$$\mathbb{E}[Z] = 0 \quad \text{and} \quad \mathbb{P}(|Z| > t) \leq 2\exp\left(-t^2/(2\sigma^2)\right) \quad \text{for } t \geq 0, \tag{A.1}$$

for some constant $\sigma > 0$.

In the proof of Theorem 1, we will rely on the following result from van de Geer (1990) (which is derived closely from Dudley's chaining for sub-Gaussian processes).

**Theorem A.1** (**Theorem 3.3 of van de Geer 1990**). *Assume that $\epsilon = (\epsilon_1, \ldots, \epsilon_n) \in \mathbb{R}^n$ has i.i.d. components drawn from a sub-Gaussian distribution, as in (A.1). Consider a set $\mathcal{X} \subseteq \mathbb{R}^n$ such that $\|x\|_n \leq 1$ for all $x \in \mathcal{X}$, and let $\mathcal{K}(\cdot)$ be a continuous function upper bounding the $\|\cdot\|_n$-entropy of $\mathcal{X}$, i.e., $\mathcal{K}(r) \geq \log N(r, \mathcal{X}, \|\cdot\|_n)$. Then there are constants $C_1, C_2, C_3, C_4 > 0$ depending only on $\sigma$ (the parameter in the underlying sub-Gaussian distribution) such that for all $t > C_1$, with*

$$t > C_2 \int_0^{t_0} \sqrt{\mathcal{K}(r)}\, dr,$$

*where $t_0 = \inf\{r : \mathcal{K}(r) \leq C_3 t^2\}$, we have*

$$\mathbb{P}\left(\sup_{x \in \mathcal{X}} \frac{|\epsilon^\top x|}{\sqrt{n}} > t\right) \leq 2\exp(-C_4 t^2).$$

Now we give the proof of Theorem 1.

*Proof of Theorem 1.* We define three events that will be critical to our proof, and we will show later on that each event occurs with high probability:

$$\Omega_0 = \left\{ \sup_{z \in \mathcal{M}} \frac{|\epsilon^\top z|}{\|z\|_2} \leq \gamma c_I \sqrt{(\log s_0 + \log\log n) s_0 \log n} \right\}, \tag{A.2}$$

$$\Omega_1 = \left\{ \sup_{w \in \mathcal{R}^\perp} \frac{|\epsilon^\top w|}{\|D_{-S_0} w\|_1^{1/2} \|w\|_2^{1/2}} \leq \gamma c_R (n s_0)^{1/4} \right\}, \tag{A.3}$$

$$\Omega_2 = \left\{ \sup_{\delta \in \mathcal{R}} \frac{|\epsilon^\top \delta|}{\|\delta\|_2} \leq \gamma c_S \sqrt{s_0} \right\}, \tag{A.4}$$

where $\gamma > 1$ is parameter free to vary in our analysis, $c_I, c_R > 0$ are the constants in Lemmas 2, 3, and $c_S > 0$ is a constant to be determined below. Focusing on the third event, we will lower bound its probability by applying Theorem A.1 to $\mathcal{X} = \mathcal{R} \cap \{\delta : \|\delta\|_n \leq 1\}$. Note that

$$\log N(r, \mathcal{R} \cap \{\delta : \|\delta\|_n \leq 1\}, \|\cdot\|_n) \leq (s_0 + 1) \log(3/r),$$

as $\mathcal{R}$ is $(s_0 + 1)$-dimensional, and it is well-known that in $\mathbb{R}^d$, the number of balls of radius $r$ that are needed to cover the unit ball is at most $(3/r)^d$. The quantity $t_0$ in Theorem A.1 may be taken to be $t_0 = \inf\{r : (s_0 + 1)\log(3/r) \leq C_3 C_1^2\} = 3\exp(-C_3 C_1^2/(s_0 + 1))$. The restrictions on $t$ are hence $t > C_1$, as well as

$$t > C_2 \int_0^{t_0} \sqrt{(s_0 + 1)\log(3/r)}\, dr.$$

But, writing $\mathrm{erf}(\cdot)$ for the error function,

$$C_2 \int_0^{t_0} \sqrt{(s_0 + 1)\log(3/r)}\, dr = (\sqrt{s_0 + 1}) \cdot 3 C_2 \left[ r\sqrt{\log\frac{1}{r}} - \frac{1}{2}\mathrm{erf}\left(\sqrt{\log\frac{1}{r}}\right) \right]\Big|_0^{t_0/3} \leq C_2 \sqrt{s_0},$$

where the constant $C_2 > 0$ is adjusted to be larger, as needed. Let us define $c_S = \max\{C_1, C_2\}$ and $C_S = C_4$. Then we have by Theorem A.1, for $t = \gamma c_S \sqrt{s_0}$ and any $\gamma > 1$,

$$1 - 2\exp(-C_S \gamma^2 c_S^2 s_0) \leq \mathbb{P}\left( \sup_{\delta \in \mathcal{R}} \frac{|\epsilon^\top \delta|}{\sqrt{n}\|\delta\|_n} \leq \gamma c_S \sqrt{s_0} \right) = \mathbb{P}\left( \sup_{\delta \in \mathcal{R}} \frac{|\epsilon^\top \delta|}{\|\delta\|_2} \leq \gamma c_S \sqrt{s_0} \right) = \mathbb{P}(\Omega_2). \tag{A.5}$$

The rest of this proof is divided into subparts for readability.

**Basic inequality.** The basic inequality in (12) is established by comparing objective values in (3) at $\widehat{\theta}$ and $\theta_0$, writing $y = \theta_0 + \epsilon$, and rearranging. Using $\widehat{\theta} - \theta_0 = P_0(\widehat{\theta} - \theta_0) + P_1(\widehat{\theta} - \theta_0) = \widehat{\delta} + \widehat{x}$, and using the fact that $\widehat{\delta}$ and $\widehat{x}$ are orthogonal, we have

$$
\begin{aligned}
\|\widehat{\delta}\|_2^2 + \|\widehat{x}\|_2^2 &\leq 2\epsilon^\top \widehat{\delta} + 2\epsilon^\top \widehat{x} + 2\lambda\big( \|D\theta_0\|_1 - \|D\widehat{\theta}\|_1 \big) \\
&= 2\epsilon^\top \widehat{\delta} + 2\epsilon^\top \widehat{x} + 2\lambda\big( \|D_{S_0}\theta_0\|_1 - \|D_{S_0}\widehat{\theta}\|_1 - \|D_{-S_0}\widehat{\theta}\|_1 \big) \\
&\leq 2\epsilon^\top \widehat{\delta} + 2\epsilon^\top \widehat{x} + 2\lambda\big( \|D_{S_0}(\theta_0 - \widehat{\theta})\|_1 - \|D_{-S_0}\widehat{\theta}\|_1 \big) \\
&\leq 2\epsilon^\top \widehat{\delta} + 2\epsilon^\top \widehat{x} + 2\lambda\big( \|D_{S_0}\widehat{\delta}\|_1 + \|D_{S_0}\widehat{x}\|_1 - \|D_{-S_0}\widehat{x}\|_1 \big) \\
&= \underbrace{2\epsilon^\top \widehat{\delta} + 2\lambda\|D_{S_0}\widehat{\delta}\|_1}_{A_0} + \underbrace{2\epsilon^\top \widehat{x} + 2\lambda\big( \|D_{S_0}\widehat{x}\|_1 - \|D_{-S_0}\widehat{x}\|_1 \big)}_{B_0},
\end{aligned}
$$

where in the third line, we used the triangle inequality, and in the fourth, we again used the triangle inequality and the fact that $D_{-S_0}\widehat{\delta} = 0$.

**Bounding $A_0$.** Note that

$$A_0 = 2\left( \frac{|\epsilon^\top \widehat{\delta}|}{\|\widehat{\delta}\|_2} + \lambda \frac{\|D_{S_0}\widehat{\delta}\|_1}{\|\widehat{\delta}\|_2} \right) \|\delta\|_2,$$

and observe

$$\|D_{S_0}\widehat{\delta}\|_1 = \sum_{i=1}^{s_0} |\widehat{\delta}_{t_{i+1}} - \widehat{\delta}_{t_i}| \leq 2\sum_{i=1}^{s_0+1} |\widehat{\delta}_{t_i}| \leq 2\sqrt{(s_0+1)\sum_{i=1}^{s_0+1}\widehat{\delta}_{t_i}^2}$$

$$\leq 4\sqrt{s_0\sum_{i=1}^{s_0+1}\frac{t_i - t_{i-1}}{W_n}\widehat{\delta}_{t_i}^2} = 4\sqrt{\frac{s_0}{W_n}}\|\widehat{\delta}\|_2.$$

The second inequality used Cauchy-Schwartz, and the last equality used that $\widehat{\delta}$ is piecewise constant on the blocks $B_0, \ldots, B_{s_0}$, as $\widehat{\delta} \in \mathcal{R} = \mathrm{span}\{\mathbb{1}_{B_0}, \ldots, \mathbb{1}_{B_{s_0}}\}$. Hence, on the event $\Omega_2$ in (A.4), we have

$$A_0 \leq 2\left(\gamma c_S\sqrt{s_0} + 4\lambda\sqrt{\frac{s_0}{W_n}}\right)\|\widehat{\delta}\|_2 \tag{A.6}$$

**Bounding $B_0$.** In the definition of $B_0$, let us expand $\widehat{x} = \widehat{z} + \widehat{w}$, where $\widehat{z} \in \mathcal{M}$ is the lower interpolant to $\widehat{x}$, as defined in Lemma 1, and $\widehat{w} = \widehat{x} - \widehat{z}$ is the remainder. Using properties (13) and (14) from Lemma 1, we arrive at

$$B_0 = 2\epsilon^\top\widehat{z} + 2\epsilon^\top\widehat{w} + 2\lambda\big(\|D_{S_0}\widehat{z}\|_1 - \|D_{-S_0}\widehat{z}\|_1 - \|D_{-S_0}\widehat{w}\|_1\big)$$

$$\leq 2\epsilon^\top\widehat{z} + 8\lambda\sqrt{\frac{s_0}{W_n}}\|\widehat{z}\|_2 + 2\epsilon^\top\widehat{w} - 2\lambda\|D_{-S_0}\widehat{w}\|_1. \tag{A.7}$$

On the event $\Omega_0$ in (A.2),

$$\epsilon^\top\widehat{z} \leq \gamma c_I\sqrt{(\log s_0 + \log\log n)s_0\log n}\|\widehat{z}\|_2.$$

And, on the event $\Omega_1$ in (A.3), as $P_1\widehat{w} \in \mathcal{R}^\perp$, $\|D_{-S_0}P_1\widehat{w}\|_1 = \|D_{-S_0}\widehat{w}\|_1$, and $\|P_1\widehat{w}\|_2 \leq \|\widehat{w}\|_2$,

$$\epsilon^\top P_1\widehat{w} \leq \gamma c_R(ns_0)^{1/4}\|D_{-S_0}\widehat{w}\|_1^{1/2}\|\widehat{w}\|_2^{1/2},$$

Also, on the event $\Omega_2$ in (A.4), since $P_0\widehat{w} \in \mathcal{R}$,

$$\epsilon^\top P_0\widehat{w} \leq \gamma c_S\sqrt{s_0}\|\widehat{w}\|_2.$$

Hence, on $\Omega_0 \cap \Omega_1 \cap \Omega_2$, combining the last three displays with (A.7),

$$B_0 \leq 2\left(\gamma c_I\sqrt{(\log s_0 + \log\log n)s_0\log n} + 4\lambda\sqrt{\frac{s_0}{W_n}}\right)\|\widehat{z}\|_2 + 2\gamma c_S\sqrt{s_0}\|\widehat{w}\|_2 +$$

$$2\gamma c_R(ns_0)^{1/4}\|D_{-S_0}\widehat{w}\|_1^{1/2}\|\widehat{w}\|_2^{1/2} - 2\lambda\|D_{-S_0}\widehat{w}\|_1. \tag{A.8}$$

Consider the first case in which $\gamma c_R(ns_0)^{1/4}\|D_{-S_0}\widehat{w}\|_1^{1/2}\|\widehat{w}\|_2^{1/2} \geq \lambda\|D_{-S_0}\widehat{w}\|_1$. Then

$$\|D_{-S_0}\widehat{w}\|_1 \leq \left(\frac{\gamma c_R}{\lambda}\right)^2\sqrt{ns_0}\|\widehat{w}\|_2,$$

and from (A.8), on the event $\Omega_0 \cap \Omega_1 \cap \Omega_2$,

$$B_0 \leq 2\left(\gamma c_I\sqrt{(\log s_0 + \log\log n)s_0\log n} + 4\lambda\sqrt{\frac{s_0}{W_n}} + \gamma c_S\sqrt{s_0} + \frac{\gamma^2 c_R^2\sqrt{ns_0}}{\lambda}\right)\|\widehat{x}\|_2. \tag{A.9}$$

where we have used (15). In the case $\gamma c_R(ns_0)^{1/4}\|D_{-S_0}\widehat{w}\|_1^{1/2}\|\widehat{w}\|_2^{1/2} < \lambda\|D_{-S_0}\widehat{w}\|_1$, we have from (A.8), on the event $\Omega_0 \cap \Omega_1 \cap \Omega_2$,

$$B_0 \leq 2\left(\gamma c_I\sqrt{(\log s_0 + \log\log n)s_0\log n} + 4\lambda\sqrt{\frac{s_0}{W_n}} + \gamma c_S\sqrt{s_0}\right)\|\widehat{x}\|_2.$$

Therefore, the bound (A.9) always holds on the event $\Omega_0 \cap \Omega_1 \cap \Omega_2$.

**Putting it all together.** Combining (A.6) and (A.9), we see that on $\Omega_0 \cap \Omega_1 \cap \Omega_2$,

$$\|\widehat{\delta}\|_2^2 + \|\widehat{x}\|_2^2 \leq 2\left(\gamma c_S\sqrt{s_0} + 4\lambda\sqrt{\frac{s_0}{W_n}}\right)\|\widehat{\delta}\|_2 +$$

$$2\left(\gamma c_I\sqrt{(\log s_0 + \log\log n)s_0\log n} + 4\lambda\sqrt{\frac{s_0}{W_n}} + \gamma c_S\sqrt{s_0} + \frac{\gamma^2 c_R^2\sqrt{ns_0}}{\lambda}\right)\|\widehat{x}\|_2.$$

Denote the right-hand side by $A_1\|\widehat{\delta}\|_2 + B_1\|\widehat{x}\|_2$. Using the simple inequality $2ab \le a^2 + b^2$, twice, we have on $\Omega_0 \cap \Omega_1 \cap \Omega_2$,

$$\|\widehat{\delta}\|_2^2 + \|\widehat{x}\|_2^2 \le \frac{A_1^2}{2} + \frac{\|\widehat{\delta}\|_2^2}{2} + \frac{B_1^2}{2} + \frac{\|\widehat{x}\|_2^2}{2}.$$

Recalling that $\|\widehat{\delta}\|_2^2 + \|\widehat{x}\|_2^2 = \|\widehat{\theta} - \theta_0\|_2^2$, this implies that on the event $\Omega_0 \cap \Omega_1 \cap \Omega_2$, there exists a constant $c > 0$, such that for large enough $n$, and any $\gamma > 1$,

$$\|\widehat{\theta} - \theta_0\|_2^2 \le \gamma^4 c s_0 \left( (\log s_0 + \log\log n)\log n + \frac{\lambda^2}{W_n} + \frac{n}{\lambda^2} \right), \tag{A.10}$$

on the event $\Omega_0 \cap \Omega_1 \cap \Omega_2$. Furthermore, using the union bound along with Lemmas 2, 3, and (A.5), we find that

$$\mathbb{P}\big((\Omega_0 \cap \Omega_1 \cap \Omega_2)^c\big) \le 2\exp\big(-C_I\gamma^2 c_I^2(\log s_0 + \log\log n)\big) +$$
$$2\exp(-C_R\gamma^2 c_R^2\sqrt{s_0}) + 2\exp(-C_S\gamma^2 c_S^2 s_0) \le \exp(-C\gamma^2),$$

for an appropriately defined constant $C > 0$. Optimizing the bound in (A.10) to choose the tuning parameter $\lambda$ yields $\lambda = (nW_n)^{1/4}$. Plugging this in gives the final result. $\qquad\square$

Next we give the proof of Corollary 1.

*Proof of Corollary 1.* Define the random variable

$$Z = \frac{\|\widehat{\theta} - \theta_0\|_2^2}{cs_0((\log s_0 + \log\log n)\log n + \sqrt{n/W_n})},$$

which we know has the tail bound $\mathbb{P}(Z > z) \le \exp(-C\sqrt{z})$ for $z > 1$, and observe that

$$\mathbb{E}(Z) = \int_0^\infty \mathbb{P}(Z > z)\, dz \le 1 + \int_1^\infty \exp(-C\sqrt{z})\, dz.$$

The right-hand side is a finite constant, and this gives the result

$$\mathbb{E}\|\widehat{\theta} - \theta_0\|_n^2 \le c\frac{s_0}{n}\left( (\log s_0 + \log\log n)\log n + \sqrt{\frac{n}{W_n}} \right),$$

where the constant $c > 0$ is adjusted to be larger, as needed. $\qquad\square$

## A.2  Proofs of Lemma 1, Lemma 2, Lemma 3, and (A.17)

*Proof of Lemma 1.* We give an explicit construction of a lower interpolant $z \in \mathcal{M}$ to $x$, given the changepoints $0 = t_0 < \ldots < t_{s_0+1} = n$. We will use the notation $a_+ = \max\{0, a\}$ for the positive part of $a$. For $i = 0, \ldots, s_0$, define $z^{(i+)} \in \mathbb{R}^{t_{i+1}-t_i}$ by setting $g_i^+ = \text{sign}(x_{t_i})$ and

$$z_j^{(i+)} = g_i^+ \cdot \min\left\{ (g_i^+ x_{t_i+1})_+, \ldots, (g_i^+ x_{t_i+j})_+ \right\}, \quad j = 1, \ldots, t_{i+1} - t_i.$$

Similarly, define $z^{(i-)} \in \mathbb{R}^{t_{i+1}-t_i}$ by setting $g_i^- = \text{sign}(x_{t_{i+1}-1})$ and

$$z_j^{(i-)} = g_i^- \cdot \min\left\{ (g_i^- x_{t_i+j})_+, \ldots, (g_i^- x_{t_{i+1}})_+ \right\}, \quad j = 1, \ldots, t_{i+1} - t_i.$$

Note that $z_1^{(i+)} = x_{t_i+1}$ and $z_{t_{i+1}-t_i}^{(i-)} = x_{t_{i+1}}$; also, $\{|z_j^{(i+)}|\}_{j=1}^{t_{i+1}-t_i}$ is a nonincreasing sequence, and $\{|z_j^{(i-)}|\}_{j=1}^{t_{i+1}-t_i}$ is nondecreasing. Furthermore,

$$\text{sign}\big(z_1^{(i+)}\big) \cdot \text{sign}\big(z_j^{(i+)}\big) \ge 0 \quad \text{and} \quad \text{sign}\big(z_{t_{i+1}-t_i}^{(i-)}\big) \cdot \text{sign}\big(z_j^{(i-)}\big) \ge 0, \quad j = 1, \ldots, t_{i+1} - t_i.$$

Lastly, notice that there exists a point $j' \in 1, \ldots, t_{i+1} - t_i - 1$ (not necessarily unique) such that

$$\min_{k \in \{1,\ldots,t_{i+1}-t_i\}} |z_k^{(i+)}| = |z_{j'+1}^{(i+)}| = |z_j^{(i+)}|, \quad j = j' + 1, \ldots, t_{i+1} - t_i, \tag{A.11}$$

$$\min_{k \in \{1,\ldots,t_{i+1}-t_i\}} |z_k^{(i-)}| = |z_{j'}^{(i-)}| = |z_j^{(i-)}|, \quad j = 1, \ldots, j'. \tag{A.12}$$

We define $z_{t_i+j} = z_j^{(i+)}$ for $j = 1, \ldots, j'$, and $z_{t_i+j} = z_j^{(i-)}$ for $j = j'+1, \ldots, t_{i+1} - t_i$. Letting $t_i' = t_i + j'$ and repeating this process for $i = 0, \ldots, s_0$, we have constructed $z \in \mathcal{M}$.

We now verify the claimed properties for the constructed lower interpolant $z$. For $i = 0, \ldots, s_0$, and any $j = 1, \ldots, t_{i+1} - t_i$, we have

$$\text{sign}(z_j^{(i+)}) \cdot \text{sign}(x_{t_i+j}) \geq 0, \tag{A.13}$$

$$|z_j^{(i+)}| \leq |x_{t_i+j}|, \tag{A.14}$$

Further, for any $j = 1, \ldots, t_{i+1} - t_i - 1$,

$$\text{sign}\left((Dz^{(i+)})_j\right) \cdot \text{sign}\left((Dx)_{t_i+j}\right) \geq 0, \tag{A.15}$$

$$\left|(Dz^{(i+)})_j\right| \leq \left|(Dx)_{t_i+j}\right|. \tag{A.16}$$

To see why (A.15) holds, note that the properties $\text{sign}(Dz^{(i+)})_j \in \{-1, 0\}$, $(Dz^{(i+)})_j < 0$ imply $(D(g_i^+ x)_+)_{t_i+j} < 0$. To see why (A.16) holds, if $(Dz^{(i+)})_j \neq 0$, then we know that

$$|z_{j+1}^{(i+)} - z_j^{(i+)}| \leq \left| \min\left\{ (g_i^+ x_{t_i+j+1})_+, (g_i^+ x_{t_i+j})_+ \right\} - (g_i^+ x_{t_i+j})_+ \right| \leq |x_{t_i+j+1} - x_{t_i+j}|,$$

where we used the observation that $|\min\{a,b\} - b| \geq |\min\{a,b,c\} - \min\{b,c\}|$.

It can be shown by nearly equivalent steps that $z^{(i-)}$, $z$ both satisfy properties analogous to (A.13)–(A.16). Using (A.13) and (A.14) on $z$ gives (15). Using (A.15) and (A.16) on $z$ gives (13) (note that if $\text{sign}(a) = \text{sign}(b)$ and $|a| > |b|$, then $|a| = |b| + |a-b|$). Because $z_{t_{i+1}} = x_{t_{i+1}}$ and $z_{t_{i+1}} = x_{t_{i+1}}$ for all $i = 0, \ldots, s_0$, we have the equality in (14) (as $D_{t_i} z = z_{t_{i+1}} - z_{t_i} = x_{t_{i+1}} - x_{t_i} = D_{t_i} x$).

Finally, for each $i = 0, \ldots, s_0$, define $t_i'' = t_i'$ if $|z_{t_i'}| \geq |z_{t_i'+1}|$ and $t_i'' = t_i' + 1$ otherwise. Observe that by (A.11) and (A.12), it holds that $|z_{t_i''}| = \min_{j=1,\ldots,t_{i+1}-t_i} |z_{t_i+j}|$. The inequality in (14) is finally established by the following chain of inequalities:

$$\|D_{S_0} z\|_1 = \sum_{i=1}^{s_0} |z_{t_i+1} - z_{t_i}| \leq \sum_{i=1}^{s_0} |z_{t_i+1}| + |z_{t_i}|$$

$$= \sum_{i=1}^{s_0} \left(|z_{t_i+1}| - |z_{t_i''}|\right) + \left(|z_{t_i}| - |z_{t_{i-1}''}|\right) + |z_{t_{i-1}''}| + |z_{t_i''}|$$

$$\leq \|D_{-S_0} z\|_1 + 2\sum_{i=0}^{s_0} |z_{t_i''}| \leq \|D_{-S_0} z\|_1 + 4\sqrt{\frac{s_0}{W_n}} \|z\|_2,$$

where in the second inequality, we used $|a| - |c| \leq |a - c| \leq |a - b| + |b - c|$, and in the last inequality, we used the above property of $z_{t_i''}$ and

$$\sum_{i=0}^{s_0} |z_{t_i''}| \leq 2\sqrt{s_0} \sqrt{\sum_{i=0}^{s_0} |z_{t_i''}|^2} \leq 2\sqrt{s_0 \sum_{i=0}^{s_0} \frac{t_{i+1} - t_i}{W_n} z_{t_i''}^2} \leq 2\sqrt{\frac{s_0}{W_n}} \|z\|_2.$$

This completes the proof. $\qquad \square$

*Proof of Lemma 2.* We consider $\epsilon \in \mathbb{R}^n$, an i.i.d. sub-Gaussian vector as referred to in the statement of the lemma, and arbitrary $z \in \mathcal{M}$. In this proof, we will also consider $E(t)$ and $Z(t)$, real-valued functions over $[0, n]$, constructed so that $E(t) = \epsilon_{\lceil t \rceil}$ for all $t$ (i.e., $E(t)$ is a step function), $Z(t) = z_t$ for $t = 1, \ldots, n$, and $Z(t)$ is continuously differentiable and monotone over $(t_i, t_i']$ and $(t_i', t_{i+1}]$ for $i = 0, \ldots, s_0$. These functions will also satisfy the boundary conditions $E(0) = \epsilon_1$ and $Z(0) = z_1$.

Let $F(t) = \int_0^t E(u)\, du$. As $\epsilon$ is random, $E(t)$ and $F(t)$ are also random. It can be shown that there exists constants $c_I, C_I > 0$ such that for any $\gamma > 1$,

$$\mathbb{P}\left( \frac{|F(t) - F(t_i)|}{\sqrt{|t - t_i|}} \leq \gamma c_I \sqrt{\log s_0 + \log\log n}, \text{ for } t \in (t_i, t_{i+1}], i = 0, \ldots, s_0 \right)$$

$$\geq 1 - 2\exp\left(-C_I \gamma^2 c_I^2 (\log s_0 + \log\log n)\right). \tag{A.17}$$

So as not to distract from the main flow of ideas, we now proceed to prove Lemma 2, and we provide a proof of (A.17) later. Let $\Omega_3$ denote the event in consideration on the left-hand side of (A.17). By integration by parts,

$$\int_{t_i}^{t_i'} E(t)Z(t)\,dt = Z(t_i')(F(t_i') - F(t_i)) - \int_{t_i}^{t_i'} Z'(t)(F(t) - F(t_i))\,dt$$

where $Z'(t) = \frac{d}{dt}Z(t)$. Thus, on the event $\Omega_3$,

$$\left| \int_{t_i}^{t_i'} E(t)Z(t)\,dt \right| \leq \gamma c_I \sqrt{\log s_0 + \log\log n} \left( |Z(t_i')|\sqrt{t_i' - t_i} + \left| \int_{t_i}^{t_i'} Z'(t)\sqrt{t - t_i}\,dt \right| \right),$$
(A.18)

since $Z'$ does not change sign within the intervals $(t_i, t_i'], (t_i', t_{i+1}]$ (as $z \in \mathcal{M}$). For $n$ large enough, we can upper bound the last term in (A.18) as follows

$$\left| \int_{t_i}^{t_i'} Z'(t)\sqrt{t - t_i}\,dt \right| \leq \left| \int_{t_i}^{t_i+n^{-1}} Z'(t)\sqrt{t - t_i}\,dt \right| + \left| \int_{t_i+n^{-1}}^{t_i'} Z'(t)\sqrt{t - t_i}\,dt \right|. \quad (A.19)$$

Using integration by parts and the triangle inequality on the second term in (A.19),

$$\left| \int_{t_i+n^{-1}}^{t_i'} Z'(t)\sqrt{t - t_i}\,dt \right| \leq |Z(t_i')|\sqrt{t_i' - t_i} + \left| \frac{Z(t_i + n^{-1})}{\sqrt{n}} \right| + \frac{1}{2}\left| \int_{t_i+n^{-1}}^{t_i'} \frac{Z(t)}{\sqrt{t - t_i}}\,dt \right|. \quad (A.20)$$

By Cauchy-Schwartz on the last term in (A.20),

$$\left| \int_{t_i+n^{-1}}^{t_i'} \frac{Z(t)}{\sqrt{t - t_i}}\,dt \right| \leq \left( \int_{t_i+n^{-1}}^{t_i'} Z(t)^2\,dt \right)^{1/2} \left( \int_{t_i+n^{-1}}^{t_i'} \frac{1}{t - t_i}\,dt \right)^{1/2}$$

$$\leq \left( \int_{t_i+n^{-1}}^{t_i'} Z(t)^2\,dt \right)^{1/2} \sqrt{2\log n}. \quad (A.21)$$

Now examining the first term in (A.19),

$$\left| \int_{t_i}^{t_i+n^{-1}} Z'(t)\sqrt{t - t_i}\,dt \right| \leq n^{-1/2}\left| \int_{t_i}^{t_i+n^{-1}} Z'(t)\,dt \right| = \frac{|Z(t_i + n^{-1}) - Z(t_i)|}{\sqrt{n}}.$$

But because we only require $Z$ to be piecewise monotonic and continuously differentiable then we are at liberty to make $Z(t_i + n^{-1}) = Z(t_i)$, forcing this term to be 0. In order to bound $Z(t_i')$, notice that because $|Z(t)|$ is non-increasing over the interval $(t_i, t_i']$ we have that

$$Z(t_i')^2|t_i' - t_i| \leq \int_{t_i}^{t_i'} Z(t)^2\,dt. \quad (A.22)$$

Combining (A.18)–(A.22), we have that on the event $\Omega_3$,

$$\left| \int_{t_i}^{t_i'} E(t)Z(t)\,dt \right| \leq \alpha_n \left( 2 + \sqrt{\frac{\log n}{2}} \right) \left( \int_{t_i}^{t_i'} Z(t)^2\,dt \right)^{1/2} + \alpha_n \frac{|Z(t_i)|}{\sqrt{n}}. \quad (A.23)$$

where we have abbreviated $\alpha_n = \gamma c_I \sqrt{\log s_0 + \log\log n}$. Through nearly identical steps we can show that on the event $\Omega_3$,

$$\left| \int_{t_i'}^{t_{i+1}} E(t)Z(t)\,dt \right| \leq \alpha_n \left( 2 + \sqrt{\frac{\log n}{2}} \right) \left( \int_{t_i'}^{t_{i+1}} Z(t)^2\,dt \right)^{1/2} + \alpha_n \frac{|Z(t_{i+1})|}{\sqrt{n}}. \quad (A.24)$$

Therefore

$$\left| \int_0^n E(t)Z(t)\,dt \right| \leq \sum_{i=0}^{s_0} \left( \left| \int_{t_i}^{t_i'} E(t)Z(t)\,dt \right| + \left| \int_{t_i'}^{t_{i+1}} E(t)Z(t)\,dt \right| \right)$$

$$\leq \alpha_n \sqrt{2s_0 + 2}\left( 2 + \sqrt{\frac{\log n}{2}} \right) \left( \int_0^n Z(t)^2\,dt \right)^{1/2} + 2\alpha_n \frac{\|z\|_1}{\sqrt{n}}, \quad (A.25)$$

where in the second line we used (A.23), (A.24), and the Cauchy-Schwartz inequality. Because we can choose $Z(t)$ to be arbitrarily close to $z_{\lceil t \rceil}$ over all $t$, we can make the integral $(\int_0^n Z(t)^2\,dt)^{1/2}$ arbitrarily close to $\|z\|_2$ and likewise we can make $\int_0^n E(t)Z(t)\,dt$ arbitrarily close to $\epsilon^\top z$. Furthermore, because $\|z\|_1 \le \sqrt{n}\|z\|_2$, the first term in (A.25) dominates. Hence on the event $\Omega_3$, we have established that

$$|\epsilon^\top z| \le \gamma c_I \sqrt{(\log s_0 + \log\log n)s_0 \log n}\,\|z\|_2,$$

where the constant $c_I$ is adjusted to be larger, as needed. Noting that the event $\Omega_3$ does not depend on $z$, the result follows. $\qquad\square$

*Proof of claim* (A.17). We will construct a covering for $\mathcal{V} = \cup_{i=0}^{s_0}\mathcal{V}_i$, where for each $i = 0, \ldots, s_0$,

$$\mathcal{V}_i = \left\{ \sqrt{\frac{n}{|A|}}\mathbb{1}_A : A = \{t_i, \ldots, t\},\ t = t_i + 1, \ldots, n \right\} \cup$$

$$\left\{ \sqrt{\frac{n}{|A|}}\mathbb{1}_A : A = \{t, \ldots, t_i\},\ t = 1, \ldots, t_i - 1 \right\}.$$

Our scaling is such that, for any $a = \sqrt{n/|A|}\mathbb{1}_A$, where $A \subseteq \{1, \ldots, n\}$, we have $\|a\|_n = 1$. Further, for any other $b = \sqrt{n/|B|}\mathbb{1}_B$, where $B \subseteq \{1, \ldots, n\}$, we have

$$\|a - b\|_n^2 = \frac{|A \cap B|}{(\sqrt{|A|} - \sqrt{|B|})^2} + \frac{|A \setminus B|}{|A|} + \frac{|B \setminus A|}{|B|} = 2\left(1 - \frac{|A \cap B|}{\sqrt{|A||B|}}\right). \tag{A.26}$$

We first construct a covering for each set $\mathcal{V}_i$, $i = 0, \ldots, s_0$, and we restrict our attention to a radius $0 < r < \sqrt{2}$. Let $\alpha = \lceil (1 - r^2/2)^{-2} \rceil$, and consider the set

$$\mathcal{C}_i = \left\{ \sqrt{\frac{n}{|A|}}\mathbb{1}_A : A = \{t_i, \ldots, \min\{t_i + \alpha^j, n\}\},\ j = 1, \ldots, \lceil \log n/\log\alpha \rceil \right\} \cup$$

$$\left\{ \sqrt{\frac{n}{|A|}}\mathbb{1}_A : A = \{\max\{t_i - \alpha^j, 1\}, \ldots, t_i\},\ j = 1, \ldots, \lceil \log n/\log\alpha \rceil \right\}.$$

Here, the set $\mathcal{C}_i$ has at most $2\lceil \log n/\log\alpha \rceil \le 4\log n/\log\alpha$ elements, and by (A.26), balls of radius $r$ around elements in $\mathcal{C}_i$ cover the set $\mathcal{V}_i$. This establishes that

$$N(r, \mathcal{V}_i, \|\cdot\|_n) \le \frac{-2\log n}{\log(1 - r^2/2)}. \tag{A.27}$$

For a radius $0 < r < \sqrt{2}$, the covering number for $\mathcal{V} = \cup_{i=0}^{s_0}\mathcal{V}_i$ can be obtained by simply taking a union of the covers in (A.27) over $i = 0, \ldots, s_0$, giving

$$N(r, \mathcal{V}, \|\cdot\|_n) \le \sum_{i=0}^{s_0} N(r, \mathcal{V}_i, \|\cdot\|_n) \le 2(s_0 + 1)\left(\frac{-\log n}{\log(1 - r^2/2)}\right). \tag{A.28}$$

Using (A.26) once more, the diameter of the set $\mathcal{V}$ is $\sqrt{2}$, hence if $r \ge 1/\sqrt{2}$, then we need only 1 ball to cover $\mathcal{V}$. Combining this fact with (A.28), we obtain

$$N(r, \mathcal{V}, \|\cdot\|_n) \le \begin{cases} 2(s_0 + 1)\left(\dfrac{-\log n}{\log(1 - r^2/2)}\right) & \text{if } 0 < r < 1/\sqrt{2} \\ 1 & \text{if } r \ge 1/\sqrt{2} \end{cases}. \tag{A.29}$$

Now let us apply Theorem A.1, with $\mathcal{X} = \mathcal{V}$. First, we remark that the quantity $t_0$ in Theorem A.1 may be taken to be $t_0 = 1/\sqrt{2}$. The bounds on $t$ in the theorem are $t > C_1$, as well as

$$t > C_2 \int_0^{1/\sqrt{2}} \sqrt{\log\left(2(s_0 + 1)\frac{-\log n}{\log(1 - r^2/2)}\right)}\,dr.$$

Next, we know that the right-hand side above is upper bounded by

$$C_2 \int_0^{1/\sqrt{2}} \left[ \sqrt{\log\left(2(s_0 + 1)\log n\right)} + \sqrt{\log\left(\frac{-1}{\log(1 - r^2/2)}\right)} \right] dr$$

$$= C_2 \sqrt{\frac{\log\left(2(s_0 + 1)\log n\right)}{2}} + C_2\sqrt{2}\int_0^{1/2} \sqrt{\log\left(\frac{1}{\log\left(\frac{1}{1 - x^2}\right)}\right)}\,dx.$$

One can verify that the the integral in the second term above converges to a finite constant (upper bounded by 1 in fact). Thus the entire expression above is upper bounded by $C_2\sqrt{\log s_0 + \log\log n}$, where the constant $C_2 > 0$ is adjusted to be larger, as needed. Therefore, letting $c_I = \max\{C_1, C_2\}$, we may restrict our attention to $t > c_I\sqrt{\log s_0 + \log\log n}$ in Theorem A.1, and letting $C_I = C_4$, the conclusion reads, for $t = \gamma c_I$ and $\gamma > 1$,

$$\mathbb{P}\left(\sup_{a\in\mathcal{V}} \frac{\epsilon^\top a}{\sqrt{n}} > \gamma c_I\sqrt{\log s_0 + \log\log n}\right) \leq 2\exp\left(-C_I\gamma^2 c_I^2(\log s_0 + \log\log n)\right).$$

Recalling the form of $a = \sqrt{n/|A|}\mathbb{1}_A \in \mathcal{V}$, the above may be rephrased as

$$\mathbb{P}\left(\frac{\sum_{j=t_i}^\top \epsilon_j}{\sqrt{|t-t_i|}} > \gamma c_I\sqrt{\log s_0 + \log\log n}, \text{ for } t = 1,\ldots,n, i = 0,\ldots,s_0\right)$$
$$\leq 2\exp\left(-C_I\gamma^2 c_I^2(\log s_0 + \log\log n)\right). \quad \text{(A.30)}$$

Finally, consider the following event

$$\Omega_4 = \left\{\frac{|F(t) - F(t_i)|}{\sqrt{|t-t_i|}} \leq \gamma c_I\sqrt{\log s_0 + \log\log n}, \text{ for } t = 1,\ldots,n, i = 0,\ldots,s_0\right\}.$$

Recalling that $E(t) = \epsilon_{\lceil t\rceil}$ for all $t \in [0,1]$, we have $F(t) = \int_0^t E(u)\,du = \sum_{j=0}^t \epsilon_j$ for $t = 1,\ldots,n$. In (A.30), we have thus shown $\mathbb{P}(\Omega_4) \geq 1 - 2\exp(-C_I\gamma^2 c_I^2(\log s_0 + \log\log n))$. Note that $|F(t) - F(t_i)|$ is piecewise linear with knots at $t = 1,\ldots,n$ and $\sqrt{|t-t_i|}$ is concave in between these knots, so if $|F(t) - F(t_i)|/\sqrt{|t-t_i|} \leq \gamma c_I\sqrt{\log s_0 + \log\log n}$ for $t = 1,\ldots,n$, then the same bound must hold over all $t \in [0,n]$. This shows that $\Omega_4 \supseteq \Omega_3$, where $\Omega_3$ is the event in question in the left-hand side of (A.17); in other words, we have verified (A.17). $\square$

For the proof of Lemma 3, we will need the following result from van de Geer (1990).

**Lemma A.1** (**Lemma 3.5 of van de Geer 1990**). *Assume the setting of Theorem A.1, and additionally, assume that for some $\zeta \in (0,1)$ and $K > 0$,*

$$\mathcal{K}(r) \leq Kr^{-2\zeta},$$

*where, recall, $\mathcal{K}(r)$ is a continuous function upper bounding the entropy number $\log N(r, \mathcal{X}, \|\cdot\|_n)$. Then there exists constants $C_0, C_1$ depending only on $\sigma$ such that for any $t \geq C_0$,*

$$\mathbb{P}\left(\sup_{x\in\mathcal{X}} \frac{|\epsilon^\top x|}{\sqrt{n}\|x\|_n^{1-\zeta}} > t\sqrt{K}\right) \leq \exp(-C_1 t^2 K).$$

*Proof of Lemma 3.* Recall that for $i = 0,\ldots,s_0$, we let $B_i = \{t_i + 1,\ldots,t_{i+1}\}$. For $i = 0,\ldots,s_0$, also define $n_i = |B_i|$, the scaled norm $\|\cdot\|_{n_i} = \|\cdot\|_2/\sqrt{n_i}$, and

$$\mathcal{X}_i = \left\{w^{(i)} \in \mathbb{R}^{n_i} : (\mathbb{1}^{(i)})^\top w^{(i)} = 0, \|D^{(i)}w^{(i)}\|_1 \leq 1, \|w^{(i)}\|_{n_i} \leq 1\right\}.$$

Here, we write $\mathbb{1}^{(i)} \in \mathbb{R}^{n_i}$ for the vector of all 1s, and $D^{(i)} \in \mathbb{R}^{(n_i-1)\times n}$ for the difference operator, as in (6) but of smaller dimension. The set $\mathcal{X}_i$ is the discrete total variation space in $\mathbb{R}^{n_i}$, where all elements are centered and have scaled norm at most 1. From well-known results on entropy bounds for total variation spaces (e.g., from Lemma 11 and Corollary 12 of Wang et al. (2017)), we have

$$\log N(r, \mathcal{X}_i, \|\cdot\|_{n_i}) \leq \frac{C}{r},$$

for a universal constant $C > 0$. Hence we may apply Lemma A.1 with $\mathcal{X} = \mathcal{X}_i$ and $\zeta = 1/2$: for the random variable

$$M_i = \sup\left\{\frac{|\epsilon_{B_i}^\top w^{(i)}|}{\sqrt{n_i}\|w^{(i)}\|_{n_i}^{1/2}} : w^{(i)} \in \mathcal{X}_i\right\},$$

we may take $t = \gamma C_0$ in the lemma, for any $\gamma > 1$, and conclude that

$$\mathbb{P}\left(M_i > \gamma C_0\sqrt{C}\right) \leq \exp(-C_1\gamma^2 C_0^2 C).$$

Notice that we may rewrite $M_i$ as

$$M_i = \sup\left\{\frac{|\epsilon_{B_i}^\top w^{(i)}|}{n_i^{1/4}\|D^{(i)}w^{(i)}\|_1^{1/2}\|w^{(i)}\|_2^{1/2}} : w^{(i)} \in \mathbb{R}^{n_i}, \ (\mathbb{1}^{(i)})^\top w^{(i)} = 0\right\},$$

and therefore

$$\mathbb{P}\left(\sup_{w^{(i)}\in\mathbb{R}^{n_i},\,(\mathbb{1}^{(i)})^\top w^{(i)}=0}\frac{|\epsilon_{B_i}^\top w^{(i)}|}{\|D^{(i)}w^{(i)}\|_1^{1/2}\|w^{(i)}\|_2^{1/2}} > \gamma C_0\sqrt{C}n_i^{1/4}\right) \le \exp(-C_1\gamma^2C_0^2 C).$$

Using the union bound,

$$\mathbb{P}\left(\sup_{\substack{w^{(i)}\in\mathbb{R}^{n_i},\,(\mathbb{1}^{(i)})^\top w^{(i)}=0\\i=0,\dots,s_0}}\frac{|\epsilon_{B_i}^\top w^{(i)}|}{\|D^{(i)}w^{(i)}\|_1^{1/2}\|w^{(i)}\|_2^{1/2}} > \gamma C_0\sqrt{C}n_i^{1/4}\right) \le (s_0+1)\exp(-C_1\gamma^2C_0^2 C).$$

Define the constants $c_R = \max\{C_0\sqrt{C}, 1\}$ and $C_R = \max\{C_1/2, 1\}$. This ensures that we have $2C_R\gamma^2 c_R^2\sqrt{s_0} \ge \log(s_0+1)$ for any $\gamma > 1$ and any $s_0$, thus

$$\mathbb{P}\left(\sup_{\substack{w^{(i)}\in\mathbb{R}^{n_i},\,(\mathbb{1}^{(i)})^\top w^{(i)}=0\\i=0,\dots,s_0}}\frac{|\epsilon_{B_i}^\top w^{(i)}|}{\|D^{(i)}w^{(i)}\|_1^{1/2}\|w^{(i)}\|_2^{1/2}} > \gamma c_R(n_i s_0)^{1/4}\right) \le \exp(-C_R\gamma^2 c_R^2\sqrt{s_0}).$$

The proof is completed by noting the following: if $w \in \mathcal{R}^\perp$, then $(\mathbb{1}^{(i)})^\top w_{B_i} = 0$ for $i = 0, \dots, s_0$, and so on the event in consideration in the last display,

$$\begin{aligned}
|\epsilon^\top w| &\le \sum_{i=0}^{s_0}|\epsilon_{B_i}^\top w_{B_i}| \le \gamma c_R s_0^{1/4}\sum_{i=0}^{s_0}n_i^{1/4}\|D^{(i)}w_{B_i}\|_1^{1/2}\|w_{B_i}\|_2^{1/2}\\
&\le \gamma c_R s_0^{1/4}\left(\sum_{i=0}^{s_0}\|D^{(i)}w_{B_i}\|_1\right)^{1/2}\left(\sum_{i=0}^{s_0}n_i^{1/2}\|w_{B_i}\|_2\right)^{1/2}\\
&= \gamma c_R s_0^{1/4}\|D_{-S_0}w\|_1^{1/2}\left(\sum_{i=0}^{s_0}n_i^{1/2}\|w_{B_i}\|_2\right)^{1/2}\\
&\le \gamma c_R s_0^{1/4}\|D_{-S_0}w\|_1^{1/2}\left(\sum_{i=0}^{s_0}\|w_{B_i}\|_2^2\right)^{1/4}\left(\sum_{i=0}^{s_0}n_i\right)^{1/4}\\
&= \gamma c_R s_0^{1/4}\|D_{-S_0}w\|_1^{1/2}\|w\|_2^{1/2}n^{1/4},
\end{aligned}$$

by two successive uses of Cauchy-Schwartz. $\qquad\square$

## A.3 Proof of Theorem 2

Let $\widehat{\theta}$ denote the fused lasso estimate in (3), and $\widetilde{\theta} \in \mathbb{R}^n$ denote an arbitrary vector. By subgradient optimality, we know that $y - \widehat{\theta} = \lambda g$ for a subgradient $g \in \mathbb{R}^n$ of the function $x \mapsto \|Dx\|_1$ evaluated at $x = \widehat{\theta}$. Thus,

$$(y - \widehat{\theta})^\top\widehat{\theta} = \lambda\|D\widehat{\theta}\|_1.$$

Furthermore,

$$(y - \widehat{\theta})^\top\widetilde{\theta} \le \lambda\|D\widetilde{\theta}\|_1.$$

Subtracting the second to last equation from the last gives

$$(y - \widehat{\theta})^\top(\widetilde{\theta} - \widehat{\theta}) \le \lambda\big(\|D\widetilde{\theta}\|_1 - \|D\widehat{\theta}\|_1\big),$$

or

$$(\theta_0 - \widehat{\theta})^\top(\widetilde{\theta} - \widehat{\theta}) \le \epsilon^\top(\widetilde{\theta} - \widehat{\theta}) + \lambda\big(\|D\widetilde{\theta}\|_1 - \|D\widehat{\theta}\|_1\big).$$

Using the polarization identity $2a^\top b = \|a\|_2^2 + \|b\|_2^2 - \|a - b\|_2^2$ gives

$$\|\widehat{\theta} - \theta_0\|_2^2 + \|\widetilde{\theta} - \widehat{\theta}\|_2^2 - \|\theta_0 - \widetilde{\theta}\|_2^2 \leq 2\epsilon^\top(\widetilde{\theta} - \widehat{\theta}) + 2\lambda(\|D\widetilde{\theta}\|_1 - \|D\widehat{\theta}\|_1).$$

As this holds for any $\widetilde{\theta}$, we can take $\widetilde{\theta} = \theta_0(s)$ in particular, and rearrange, to find that

$$\|\widehat{\theta} - \theta_0\|_2^2 + \|\widehat{\theta} - \theta_0(s)\|_2^2 \leq \|\theta_0(s) - \theta_0\|_2^2 + 2\epsilon^\top(\theta_0(s) - \widehat{\theta}) + 2\lambda(\|D\theta_0(s)\|_1 - \|D\widehat{\theta}\|_1). \quad \text{(A.31)}$$

The right-hand side above can be handled just as in the proof of Theorem 1. Dropping $\|\theta_0(s) - \widehat{\theta}\|_2^2$ from the left-hand side above proves the first display (17) in the theorem.

To prove the second display (18) in the theorem, observe that on $E$, $\|\widehat{\theta} - \theta_0\|_2^2 \geq \|\theta_0(s) - \theta_0\|_2^2$ by construction of $\theta_0(s)$; thus from (A.31), we have

$$\|\widehat{\theta} - \theta_0(s)\|_2^2 \leq 2\epsilon^\top(\theta_0(s) - \widehat{\theta}) + 2\lambda(\|D\theta_0(s)\|_1 - \|D\widehat{\theta}\|_1).$$

and the right-hand side here can be again handled as in the proof of Theorem 1. $\qquad \square$

## A.4 Proofs of Theorem 3 and (21)

*Proof of Theorem 3.* For each $i = 1, \ldots, n$, consider the univariate negative log-likelihood function $g$ defined by

$$g_i(\theta_i) = -y_i\theta_i + \Lambda(\theta_i).$$

This is a strictly convex, twice continuously differentiable function, due to our assumptions on the cumulant generating function $\Lambda$. Therefore, the second derivative of $g$ satisfies

$$g_i''(\theta_i) = \Lambda''(\theta_i) \geq m,$$

i.e., its has a (strictly positive) minimum on the compact interval $[l, u]$, which we denote as $m > 0$. Now define $f(\theta) = \sum_{i=1}^n g(\theta_i)$ as the negative log-likelihood loss over all $n$ samples. The above display implies that

$$f(\theta) - f(\theta_0) - \nabla f(\theta_0)^\top(\theta - \theta_0) \geq \frac{m}{2}\|\theta - \theta_0\|_2^2, \quad \text{for } \theta_i, \theta_{0,i} \in [\ell, u], i = 1, \ldots, n. \quad \text{(A.32)}$$

Returning to our estimate $\widehat{\theta}$ in (20), by comparing the objectives at $\widehat{\theta}$ and at $\theta_0$, we have

$$f(\widehat{\theta}) + \lambda\|D\widehat{\theta}\|_1 \leq f(\theta_0) + \lambda\|D\theta_0\|_1.$$

Rearranging the terms in the above display and using (A.32), we have

$$\frac{m}{2}\|\widehat{\theta} - \theta_0\|_2^2 \leq -\nabla f(\theta_0)^\top(\widehat{\theta} - \theta_0) + \lambda(\|D\theta_0\|_1 - \|D\widehat{\theta}\|_1). \quad \text{(A.33)}$$

By assumption, the components of the random vector $-\nabla f(\theta_0)$, namely

$$-\nabla_i f(\theta_0) = y_i - \Lambda'(\theta_{0,i}) = y_i - \mathbb{E}(y_i), \quad i = 1, \ldots, n,$$

follow a sub-Gaussian distribution. Thus the right-hand side in (A.33) can be analyzed exactly as in the proof of Theorem 1, which leads to the desired result. $\qquad \square$

*Proof of* (21). From (A.33), observe

$$\frac{m}{2M}\|\widehat{\theta} - \theta_0\|_2^2 \leq \frac{-\nabla f(\theta_0)^\top}{M}(\widehat{\theta} - \theta_0) + \frac{\lambda}{M}(\|D\theta_0\|_1 - \|D\widehat{\theta}\|_1), \quad \text{(A.34)}$$

where $M > 1$ is a parameter free to vary, that we will specify below. Define an event

$$E = \{y : \|\nabla f(\theta_0)\|_\infty \leq M\} = \bigcap_{i=1}^n \{y_i : |y_i - \mu(\theta_{0,i})| \leq M\},$$

On $E$, the random vector $-\nabla f(\theta_0)/M$ has sub-Gaussian components (since it is bounded), and the right-hand side in (A.34) can be analyzed as in the proof of Theorem 1. The final error bound will be the usual error bound (i.e., that from Theorem 1) multiplied by a factor of $M$.

Now we bound the probability of $E$. For $W \sim \text{Pois}(\mu)$, by Poisson concentration results (Pollard, 2015),

$$\mathbb{P}(|W - \mu| > x) \leq 2\exp\left(-\frac{x^2}{2\mu}\psi\left(\frac{x}{\mu}\right)\right), \quad \text{for } x > 0, \text{ where } \psi(x) = \frac{(1+x)\log(1+x) - x}{x^2/2}.$$

$\qquad \square$

Observe for any $x \geq 1$,

$$\frac{x^2}{2\mu} \psi\left(\frac{x}{\mu}\right) \geq \frac{x^2}{2\mu} \frac{1}{1 + x/(3\mu)} \geq \frac{1/2}{\mu + 1/3} x.$$

Setting $M = \delta \log n$ for a constant $\delta > 0$ to be determined, and using the last two displays, as well as the bound $\mu(\theta_{0,i}) = e^{\theta_{0,i}} \leq e^u$, $i = 1, \ldots, n$, yields

$$\mathbb{P}(E^c) \leq n \exp\left(-\frac{1/2}{e^u + 1/3}\delta \log n\right) = \exp\left(\left(1 - \frac{1/2}{e^u + 1/3}\delta\right) \log n\right).$$

Now we simply need to choose $\delta$ large enough so that the right-hand side above equals $1/n$, i.e., we choose $\delta = 4(e^u + 1/3)$, and this completes the proof.

## A.5  Proof of Theorem 4

Fix any $\epsilon > 0$. By assumption, we know that there is a constant $C > 0$ and an integer $N_1 > 0$ such that

$$\mathbb{P}\left(\|\widetilde{\theta} - \theta_0\|_n^2 > \frac{C}{4}R_n\right) \leq \epsilon,$$

for all $n \geq N_1$. We also know that there is an integer $N_2 > 0$ such that $2CnR_n/H_n^2 \leq W_n$ for all $n \geq N_2$. Let $N = \max\{N_1, N_2\}$, take $n \geq N$, and let $r_n = \lfloor CnR_n/H_n^2 \rfloor$.

Suppose that $d(S(\widetilde{\theta}) \,|\, S_0) > r_n$. Then, by definition, there exists a changepoint $t_i \in S_0$ such that no changepoints of $\widetilde{\theta}$ are within $r_n$ of $t_i$, which means that $\widetilde{\theta}_j$ is constant over $j \in \{t_i - r_n + 1, \ldots, t_i + r_n\}$. Denote

$$z = \widetilde{\theta}_{t_i - r_n + 1} = \ldots = \widetilde{\theta}_{t_i} = \widetilde{\theta}_{t_i + 1} = \ldots = \widetilde{\theta}_{t_i + r_n}.$$

We then form the lower bound

$$\frac{1}{n} \sum_{j = t_i - r_n + 1}^{t_i + r_n} \left(\widetilde{\theta}_j - \theta_{0,j}\right)^2 = \frac{r_n}{n}\left(z - \theta_{0,t_i}\right)^2 + \frac{r_n}{n}\left(z - \theta_{0,t_i+1}\right)^2 \geq \frac{r_n H_n^2}{2n} > \frac{C}{4}R_n,$$

where the first inequality holds because $(x - a)^2 + (x - b)^2 \geq (a - b)^2/2$ for all $x$ (the quadratic in $x$ here is minimized at $x = (a + b)/2$), and the second because $r_n = \lfloor CnR_n/H_n^2 \rfloor$. Therefore, we see that $d(S(\widetilde{\theta}) \,|\, S_0) > r_n$ implies

$$\|\widetilde{\theta} - \theta_0\|_n^2 \geq \frac{1}{n} \sum_{j = t_i - r_n + 1}^{t_i + r_n} \left(\widetilde{\theta}_j - \theta_{0,j}\right)^2 > \frac{C}{4}R_n,$$

which implies

$$\mathbb{P}\left(d(S(\widetilde{\theta}) \,|\, S_0) > r_n\right) \leq \mathbb{P}\left(\|\widetilde{\theta} - \theta_0\|_n^2 > \frac{C}{4}R_n\right) \leq \epsilon,$$

for all $n \geq N$, completing the proof. $\qquad\qquad\qquad\qquad\qquad\qquad\qquad\qquad\qquad\qquad\qquad\quad$ $\square$

## A.6  Approximate changepoint recovery result, using post-processing

Here we state and prove a general result on approximate changepoint recovery using post-processing. It is a precursor to the result in Theorem 5 and will be used to prove the latter.

**Theorem A.2.** *Let $\widetilde{\theta} \in \mathbb{R}^n$ be such that $\|\widetilde{\theta} - \theta_0\|_n^2 = O_\mathbb{P}(R_n)$. Consider the following procedure: we evaluate the filter in (24) with bandwidth $b_n$ at all locations $i = b_n, \ldots, n - b_n$, and only keep the locations whose absolute filter value is greater than or equal to a threshold $\tau_n$. Denote the resulting filtered set by*

$$S_A(\widetilde{\theta}) = \left\{i \in \{b_n, \ldots, n - b_n\} : |F_i(\widetilde{\theta})| \geq \tau_n\right\}.$$

*For bandwidth and threshold values satisfying $b_n = \omega(nR_n/H_n^2)$, $2b_n \leq W_n$, and $\tau_n/H_n \to \rho \in (0, 1)$ as $n \to \infty$, we have*

$$\mathbb{P}\left(d_H\left(S_A(\widetilde{\theta}), S_0\right) \leq b_n\right) \to 1 \quad \text{as } n \to \infty.$$

*Proof.* The proof is not complicated conceptually, but requires some careful bookkeeping. Also, we make use of a few key lemmas whose details will be given later. Fix $\epsilon > 0$. Let $C > 0$ and $N_1 > 0$ be an integer such that for all $n \geq N_1$,

$$\mathbb{P}\Big(\|\widetilde{\theta} - \theta_0\|_n^2 > CR_n\Big) \leq \frac{\epsilon}{2}.$$

Set $\epsilon = \min\{\rho, 1-\rho\}/2$. As $b_n = \omega(nR_n/H_n^2)$, there is an integer $N_2 > 0$ such that for all $n \geq N_2$,

$$\frac{2CnR_n}{b_n} \leq (0.99\epsilon H_n)^2.$$

As $\tau_n/H_n \to \rho \in (0, 1)$, there is an integer $N_3 > 0$ such that for all $n \geq N_3$,

$$(\rho - \epsilon)H_n \leq \tau_n \leq (\rho + \epsilon)H_n.$$

Set $N = \max\{N_1, N_2, N_3\}$, and take $n \geq N$. Note that $\epsilon \leq \rho - \epsilon$ and $\rho + \epsilon \leq 1 - \epsilon$ by construction, and thus by the last two displays,

$$\sqrt{\frac{2CnR_n}{b_n}} < \tau_n < H_n - \sqrt{\frac{2CnR_n}{b_n}}. \tag{A.35}$$

Now observe

$$\mathbb{P}\Big(d_H\big(S_A(\widetilde{\theta}), S_0\big) > b_n\Big) \leq \mathbb{P}\Big(d\big(S_A(\widetilde{\theta}) \,|\, S_0\big) > b_n\Big) + \mathbb{P}\Big(d\big(S_0 \,|\, S_A(\widetilde{\theta})\big) > b_n\Big). \tag{A.36}$$

We focus on bounding each term on the right-hand side above separately. For the first term on the right-hand side in (A.36), observe that if $F_{t_i}(\theta) \geq \tau_n$ for all $t_i \in S_0$, then $d(S_A(\theta) \,|\, S_0) \leq b_n$. By the contrapositive,

$$\mathbb{P}\Big(d\big(S_A(\widetilde{\theta}) \,|\, S_0\big) > b_n\Big) \leq \mathbb{P}\Big(|F_{t_i}(\widetilde{\theta})| < \tau_n \text{ for some } t_i \in S_0\Big)$$

$$\leq \mathbb{P}\Big(|F_{t_i}(\widetilde{\theta})| < H_n - \sqrt{\frac{2CnR_n}{b_n}} \text{ for some } t_i \in S_0\Big), \tag{A.37}$$

where in the second line we used the upper bound on $\tau_n$ in (A.35). Suppose that $\|\widetilde{\theta} - \theta_0\|_n^2 \leq CR_n$; then, for $t_i \in S_0$, Lemma A.3 tells us how small $|F_{t_i}(\theta)|$ can be made with this error bound in place. Specifically, define

$$a = (\underbrace{-1/b_n, \ldots, -1/b_n}_{b_n \text{ times}}, \underbrace{1/b_n, \ldots, 1/b_n}_{b_n \text{ times}}) \quad \text{and} \quad c = (\theta_{0, t_i - b_n + 1}, \ldots, \theta_{0, t_i + b_n}),$$

and also $r = \sqrt{CnR_n}$. Then Lemma A.3 implies the following: if $\|\widetilde{\theta} - \theta_0\|_n^2 \leq CR_n$, then

$$|F_{t_i}(\widetilde{\theta})| \geq |a^\top c| - r\|a\|_2 \geq |\theta_{0, t_i + 1} - \theta_{0, t_i}| - \sqrt{\frac{2CnR_n}{b_n}} \geq H_n - \sqrt{\frac{2CnR_n}{b_n}}.$$

Therefore, continuing on from (A.37),

$$\mathbb{P}\Big(d\big(S_A(\widetilde{\theta}) \,|\, S_0\big) > b_n\Big) \leq \mathbb{P}\Big(|F_{t_i}(\widetilde{\theta})| < H_n - \sqrt{\frac{2CnR_n}{b_n}} \text{ for some } t_i \in S_0\Big)$$

$$\leq \mathbb{P}\Big(\|\widetilde{\theta} - \theta_0\|_n^2 > CR_n\Big)$$

$$\leq \frac{\epsilon}{2}.$$

It suffices to consider the second term in (A.36), and show that this is also bounded by $\epsilon/2$. Note that

$$\mathbb{P}\Big(d\big(S_0 \,|\, S_A(\widetilde{\theta})\big) > b_n\Big) \leq \mathbb{P}\Big(|F_i(\widetilde{\theta})| \geq \tau_n \text{ at some } i \text{ such that } \theta_{0, i - b_n + 1} = \ldots = \theta_{0, i + b_n}\Big)$$

$$\leq \mathbb{P}\Big(|F_i(\widetilde{\theta})| > \sqrt{\frac{2CnR_n}{b_n}} \text{ at some } i \text{ such that } \theta_{0, i - b_n + 1} = \ldots = \theta_{0, i + b_n}\Big). \tag{A.38}$$

In the second inequality we used the lower bound on $\tau_n$ in (A.35). Similar to the previous argument, suppose that $\|\widetilde{\theta} - \theta_0\|_n^2 \leq CR_n$; for any location $i$ in consideration in (A.38), Lemma A.2 tells us how large $|F_i(\widetilde{\theta})|$ can be made with this error bound in place. Defining

$$a = (\underbrace{-1/b_n, \ldots, -1/b_n}_{b_n \text{ times}}, \underbrace{1/b_n, \ldots, 1/b_n}_{b_n \text{ times}}) \quad \text{and} \quad c = (\theta_{0,i-b_n+1}, \ldots, \theta_{0,i+b_n}),$$

and $r = \sqrt{CnR_n}$, as before, the lemma says the following: if $\|\widetilde{\theta} - \theta_0\|_n^2 \leq CR_n$, then

$$|F_i(\widetilde{\theta})| \leq |a^\top c| + r\|a\|_2 = \sqrt{\frac{2CnR_n}{b_n}}.$$

Hence, continuing on from (A.38),

$$\mathbb{P}\Big(d\big(S_0 \mid S_A(\widetilde{\theta})\big) > b_n\Big) \leq \mathbb{P}\bigg(|F_i(\widetilde{\theta})| > \sqrt{\frac{2CnR_n}{b_n}} \text{ at some } i \text{ such that } \theta_{0,i-b_n+1} = \ldots = \theta_{0,i+b_n}\bigg)$$

$$\leq \mathbb{P}\Big(\|\widetilde{\theta} - \theta_0\|_n^2 > CR_n\Big)$$

$$\leq \frac{\epsilon}{2},$$

completing the proof. $\qquad\square$

## A.7 Lemmas A.2 and A.3

The proof of Theorem A.2 above relied on two lemmas, that we state below. Their proofs are based on simple arguments in convex analysis.

**Lemma A.2.** *Given* $a, c \in \mathbb{R}^m$, $r \geq 0$, *the optimal value of the (nonconvex) optimization problem*

$$\max_{x \in \mathbb{R}^m} \ |a^\top x| \ \text{ such that } \ \|x - c\|_2 \leq r \qquad (A.39)$$

*is* $|a^\top c| + r\|a\|_2$.

*Proof.* We first consider the convex optimization problem

$$\min_{x \in \mathbb{R}^m} \ a^\top x \ \text{ such that } \ \|x - c\|_2 \leq r, \qquad (A.40)$$

whose Lagrangian may be written as, for a dual variable $\lambda \geq 0$,

$$L(x, \lambda) = a^\top x + \lambda(\|x - c\|_2^2 - r^2).$$

The stationarity condition is $a + \lambda(x - c) = 0$, thus $x = c - a/\lambda$. By primal feasibility, $\|x - c\|_2 \leq r$, we see that we can take $\lambda = \|a\|_2/r$, which gives a solution $x = c - ra/\|a\|_2$. The optimal value in (A.40) is therefore $a^\top x = a^\top c - r\|a\|_2$. By the same logic, the optimal value of the convex problem

$$\max_{x \in \mathbb{R}^m} \ a^\top x \ \text{ such that } \ \|x - c\|_2 \leq r \qquad (A.41)$$

is $a^\top c + r\|a\|_2$. Now we can read off the optimal value of (A.39) from those of (A.40), (A.41): its optimal value is

$$\max\big\{-\big(a^\top c - r\|a\|_2\big), \ a^\top c + r\|a\|_2\big\} = |a^\top c| + r\|a\|_2,$$

completing the proof. $\qquad\square$

**Lemma A.3.** *Given* $a, c \in \mathbb{R}^m$, $r \geq 0$ *such that* $|a^\top c| - r\|a\|_2 \geq 0$, *the optimal value of the (convex) optimization problem*

$$\min_{x \in \mathbb{R}^n} \ |a^\top x| \ \text{ such that } \ \|x - c\|_2 \leq r \qquad (A.42)$$

*is* $|a^\top c| - r\|a\|_2$.

*Proof.* The proof is nearly immediate from the proof of Lemma A.2, above. Notice that the optimal value of (A.42) is lower bounded by that of (A.40), which we already know is $a^\top c - r\|a\|_2^2$. But when the latter is nonnegative, this is also the optimal value of (A.42). Repeating the argument with $-a$ in place of $a$ gives the result as stated in the lemma. $\qquad\square$

## A.8 Proof of Theorem 5

We will show that

$$\left\{ d_H\big(S_A(\widetilde{\theta}), S_0\big) \leq b_n \right\} \subseteq \left\{ d_H\big(S_F(\widetilde{\theta}), S_0\big) \leq 2b_n \right\}, \tag{A.43}$$

Since the left-hand side occurs with probability tending to 1, by Theorem A.2, so will the right-hand side. To show the desired containment, recall that, by the definition of Hausdorff distance,

$$\left\{ d_H\big(S_A(\widetilde{\theta}), S_0\big) \leq b_n \right\} = \left\{ d\big(S_0 \mid S_A(\widetilde{\theta})\big) \leq b_n \right\} \cap \left\{ d\big(S_A(\widetilde{\theta}) \mid S_0\big) \leq b_n \right\}. \tag{A.44}$$

Inspecting the first term on the right-hand side of (A.44), we observe

$$\left\{ d\big(S_0 \mid S_A(\widetilde{\theta})\big) \leq b_n \right\} \subseteq \left\{ d\big(S_0 \mid S_A(\widetilde{\theta})\big) \leq 2b_n \right\} \subseteq \left\{ d\big(S_0 \mid S_F(\widetilde{\theta})\big) \leq 2b_n \right\}, \tag{A.45}$$

where the last containment holds as $S_F(\widetilde{\theta}) \subseteq S_A(\widetilde{\theta})$. Inspecting the second term on the right-hand side of (A.44), we apply Lemma A.4 which states that for each $j \in \{b_n, \ldots, n - b_n\}$, there exists $i \in I_F(\widetilde{\theta})$ such that $|i - j| \leq b_n$ and $|F_i(\widetilde{\theta})| \geq |F_j(\widetilde{\theta})|$. Using this, we see

$$\left\{ d\big(S_A(\widetilde{\theta}) \mid S_0\big) \leq b_n \right\} = \left\{ \text{for all } \ell \in S_0, \text{ there exists } j \in S_A(\widetilde{\theta}) \text{ such that } |\ell - j| \leq b_n \right\}$$

$$\subseteq \left\{ \text{for all } \ell \in S_0, \text{ there exists } i \in I_F(\widetilde{\theta}) \text{ such that } |\ell - i| \leq 2b_n \right\}$$

$$= \left\{ d\big(S_F(\widetilde{\theta}) \mid S_0\big) \leq 2b_n \right\}. \tag{A.46}$$

Above, we have used Lemma A.4 for the containment in the second line. Combining (A.44), (A.45), and (A.46), we have established (A.43), as desired. $\qquad\square$

## A.9 Lemma A.4

The proof of Theorem 5 relied on the following lemma.

**Lemma A.4.** *Let $I_F(\widetilde{\theta})$ be the candidate set defined in Theorem 5. For each $j \in \{b_n, \ldots, n - b_n\}$ where $|F_j(\widetilde{\theta})| > 0$, there exists $i \in I_F(\widetilde{\theta})$ such that $|i - j| \leq b_n$ and $|F_i(\widetilde{\theta})| \geq |F_j(\widetilde{\theta})|$.*

*Proof.* To facilitate the proof, we define the concept of a *local maximum* among the absolute filter values: a location $i$ is a local maximum if its absolute filter value $|F_i(\widetilde{\theta})|$ is be greater than or equal to the absolute values at neighboring locations, and strictly greater than at least one of these values (where the boundary points are treated as having just one neighboring location). Specifically, a local maximum $i$ must satisfy one of the following conditions

$$|F_{i-1}(\widetilde{\theta})| < |F_i(\widetilde{\theta})|, \; |F_{i+1}(\widetilde{\theta})| \leq |F_i(\widetilde{\theta})|, \qquad \text{if } i \in \{b_n + 1, \ldots, n - b_n - 1\}, \tag{A.47}$$

$$|F_{i-1}(\widetilde{\theta})| \leq |F_i(\widetilde{\theta})|, \; |F_{i+1}(\widetilde{\theta})| < |F_i(\widetilde{\theta})|, \qquad \text{if } i \in \{b_n + 1, \ldots, n - b_n - 1\}, \tag{A.48}$$

$$|F_{i+1}(\widetilde{\theta})| < |F_i(\widetilde{\theta})| \qquad\qquad\qquad\qquad\qquad\qquad \text{if } i = b_n, \tag{A.49}$$

$$|F_{i-1}(\widetilde{\theta})| < |F_i(\widetilde{\theta})| \qquad\qquad\qquad\qquad\qquad\qquad \text{if } i = n - b_n. \tag{A.50}$$

Let $L(\widetilde{\theta})$ denote the set of local maximums derived from the filter with bandwidth $b_n$, i.e., the set of locations $i$ satisfying one of the four conditions (A.47)–(A.50).

We first establish that $L(\widetilde{\theta}) \subseteq I_F(\widetilde{\theta})$. Fix $i \in L(\widetilde{\theta})$. The boundary cases, $i = b_n$ or $i = n - b_n$, are handled directly by the definition of $I_F(\widetilde{\theta})$. Hence, we may assume that $i \in \{b_n + 1, \ldots, n - b_n - 1\}$, and without a loss of generality,

$$|F_i(\widetilde{\theta})| > |F_{i-1}(\widetilde{\theta})| \quad \text{and} \quad |F_i(\widetilde{\theta})| \geq |F_{i+1}(\widetilde{\theta})|,$$

as well as $F_i(\widetilde{\theta}) > 0$. This means that

$$F_i(\widetilde{\theta}) > |F_{i-1}(\widetilde{\theta})| \quad \text{and} \quad F_i(\widetilde{\theta}) \geq |F_{i+1}(\widetilde{\theta})|,$$

which of course implies

$$F_i(\widetilde{\theta}) > F_{i-1}(\widetilde{\theta}) \quad \text{and} \quad F_i(\widetilde{\theta}) \geq F_{i+1}(\widetilde{\theta}).$$

Applying the definition of the filter in (24) gives

$$\left(\sum_{j=i+1}^{i+b_n} \widetilde{\theta}_j - \sum_{j=i-b_n+1}^{i} \widetilde{\theta}_j\right) - \left(\sum_{j=i}^{i+b_n-1} \widetilde{\theta}_j - \sum_{j=i-b_n}^{i-1} \widetilde{\theta}_j\right) > 0$$

$$\left(\sum_{j=i+1}^{i+b_n} \widetilde{\theta}_j - \sum_{j=i-b_n+1}^{i} \widetilde{\theta}_j\right) - \left(\sum_{j=i+2}^{i+b_n+1} \widetilde{\theta}_j - \sum_{j=i-b_n+2}^{i+1} \widetilde{\theta}_j\right) \geq 0,$$

or, after simplification,

$$\widetilde{\theta}_{i+b_n} - 2\widetilde{\theta}_i + \widetilde{\theta}_{i-b_n} > 0 \quad \text{and} \quad -\widetilde{\theta}_{i+b_n+1} + 2\widetilde{\theta}_{i+1} - \widetilde{\theta}_{i-b_n+1} \geq 0.$$

Adding the above two equations together, we get

$$-\big(\widetilde{\theta}_{i+b_n+1} - \widetilde{\theta}_{i+b_n}\big) + 2\big(\widetilde{\theta}_{i+1} - \widetilde{\theta}_i\big) - \big(\widetilde{\theta}_{i-b_n+1} - \widetilde{\theta}_{i-b_n}\big) > 0,$$

which implies at least one of the three bracketed pairs of terms must be nonzero, i.e., a changepoint must occur at one of the locations $i$, $i + b_n$, or $i - b_n$. The proves that $L(\widetilde{\theta}) \subseteq I_F(\widetilde{\theta})$.

Now we show the intended statement. Let $j \in \{b_n, \dots, n - b_n\}$, and $i \in L(\widetilde{\theta})$ be in the direction of ascent from $j$ with respect to $F(\widetilde{\theta})$, where $j \leq i$, without a loss of generality (for the case $i < j$, replace $\ell + b_n$ below by $\ell - b_n$). That is, the location $i$ is a local maximum where

$$|F_j(\widetilde{\theta})| \leq |F_{j+1}(\widetilde{\theta})| \leq \dots \leq |F_{i-1}(\widetilde{\theta})| \leq |F_i(\widetilde{\theta})|. \tag{A.51}$$

If $|i - j| \leq b_n$, then we have the desired result, due to (A.51). If $|i - j| > b_n$, then there must be at least one location $\ell \in S(\widetilde{\theta})$ such that $|\ell - j| \leq b_n$. (To see this, note that if $\widetilde{\theta}_{j-b_n+1} = \dots = \widetilde{\theta}_{j+b_n}$, then $F_j(\widetilde{\theta}) = 0$.) Thus, at least one of $\ell, \ell + b_n$ lies in between $j$ and $i$, and then again (A.51) implies the result, completing the proof. □

## A.10  Comparison of Corollaries 4 and 5 to other results in the literature

Below are some remarks on the results in Corollaries 4 and 5.

**Remark A.1** (**Recovery under weak sparsity, comparison to BS**). *The weak sparsity result in* (25) *of Corollary 4 considers a challenging setting in which the number of changepoints $s_0$ in $\theta_0$ could be growing quickly with $n$, and we only have control on $C_n = \|D\theta_0\|_1$. We draw a comparison here to known results on binary segmentation (BS). The result in* (25) *on the (filtered) fused lasso and Theorem 3.1 in Fryzlewicz (2014) on the BS estimator $\widehat{\theta}^{\mathrm{BS}}$, each under the appropriate conditions on $W_n, H_n$, state that*

$$d_H\big(S_F(\widehat{\theta}), S_0\big) \leq \frac{2n^{1/3} C_n^{2/3} \log n}{H_n^2} \quad \text{vs.} \quad d_H\big(S(\widehat{\theta}^{\mathrm{BS}}), S_0\big) \leq \frac{cn \log n}{H_n^2} \quad \text{respectively,} \quad \text{(A.52)}$$

*where $c > 0$ is a constant, and both bounds hold with probability approaching 1. The result on $S_F(\widehat{\theta})$ is obtained by choosing $\nu_n = \sqrt{\log n}$ and then $b_n = \lfloor n^{1/3} C_n^{2/3} \log n / H_n^2 \rfloor$ in* (25). *Examining* (A.52)*, we see that, when $C_n$ scales more slowly than $n$, Corollary 4 provides the stronger result: the term $n^{1/3} C_n^{2/3}$ will be smaller than $n$, and hence the bound on $d_H(S_F(\widehat{\theta}), S_0)$ will be sharper than that on $d_H(S(\widehat{\theta}^{\mathrm{BS}}), S_0)$.*

*But we must also examine the specific restrictions that each result in* (A.52) *places on $s_0, W_n, H_n$. Consider the simplification $W_n = \Theta(n/s_0)$, corresponding to a case in which the changepoints in $\theta_0$ are spaced evenly apart. Corollary 4, starting with the condition $n^{1/3} C_n^{2/3} \log n / H_n^2 \leq W_n/2$, plugging in the relationship $C_n \geq s_0 H_n$, and rearranging to derive a lower bound on the minimum signal gap, requires $H_n = \Omega(s_0^{5/4} n^{-1/2} \log^{3/4} n)$. If $s_0 = \Theta(n^{2/5})$, then we see that the minimum signal gap requirement becomes $H_n = \Omega(\log^{3/4} n)$, which is growing with $n$ and is thus too stringent to be interesting (Sharpnack et al. (2012) showed simple thresholding of pairwise differences achieves perfect recovery when $H_n = \omega(\sqrt{\log n})$). Hence, to accommodate signals for which $H_n$ remains constant or even shrinks with $n$, we must restrict the number of jumps in $\theta_0$ according to $s_0 = O(n^{2/5 - \delta})$, for any fixed $\delta > 0$. Meanwhile, inspection of Assumption 3.2 in Fryzlewicz (2014) reveals that his Theorem 3.1 requires $s_0 = O(n^{1/4 - \delta})$, for any $\delta > 0$, in order to handle signals such that $H_n$ remains constant or shrinks with $n$. In short, the (effectively) allowable range for $s_0$ is*

*larger for Corollary 4 than for Theorem 3.1 in Fryzlewicz (2014). Even when we look within their common range, Corollary 4 places weaker conditions on $H_n$. As an example, consider $s_0 = \Theta(n^{1/6})$ and $W_n = \Theta(n^{5/6})$. The fused lasso result in (A.52) requires $H_n = \Omega(n^{-7/24} \log^{4/3} n)$, and the BS result in (A.52) requires $H_n = \Omega(n^{-1/6+\delta})$, for any $\delta > 0$. Finally, to reiterate, the fused lasso result in (A.52) gives a better Hausdorff recovery bound when $C_n$ is small compared to $n$; at the extreme end, this is better by a full factor of $n^{2/3}$, when $C_n = O(1)$.*

*While the post-processed fused lasso looks favorable compared to BS, based on its approximate changepoint recovery properties in the weak sparsity setting, we must be clear that the analyses for other methods—wild binary segmentation (WBS), the simultaneous multiscale changepoint estimator (SMUCE), and tail-greedy unbiased Haar (TGUH) wavelets—are still much stronger in this setting. Such methods have Hausdorff recovery bounds that are only possible for the post-processed fused lasso (at least, using our current analysis technique) when we assume strong sparsity. We discuss this next.*

**Remark A.2** (**Recovery under strong sparsity, comparison to other methods**). *When $s_0 = O(1)$ and $W_n = \Theta(n)$, the result in (26) in Corollary 4 shows that the post-processed fused lasso estimator delivers a Hausdorff bound of*

$$d_H\big(S_F(\widehat{\theta}), S_0\big) \leq \frac{2 \log^2 n}{H_n^2}, \tag{A.53}$$

*on the set $S_F(\widehat{\theta})$ of filtered changepoints, with probability approaching 1. This is obtained by choosing (say) $\nu_n = \sqrt{\log n / \log \log n}$ and $b_n = \lfloor \log^2 n / H_n^2 \rfloor \leq W_n/2$ in the corollary. The effective restriction on the minimum signal gap is thus $H_n = \Omega(\log n / \sqrt{n})$, which is quite reasonable, as $H_n = \omega(1/\sqrt{n})$ is needed for any method to detect a changepoint with probability tending to 1. Several other methods—the Potts estimator (Boysen et al., 2009), binary segmentation (BS) and wild binary segmentation (WBS) (Fryzlewicz, 2014), the simultaneous multiscale changepoint estimator (SMUCE) (Frick et al., 2014), and tail-greedy unbiased Haar wavelets (TGUH) (Fryzlewicz, 2016)— all admit Hausdorff recovery bounds that essentially match (A.53), under similarly weak restrictions on $H_n$. But, it should be noted that the latter three methods—WBS, SMUCE, and TGUH—continue to enjoy these same sharp Hausdorff bounds outside of the strong sparsity setting, i.e., their analyses do not require that $s_0 = O(1)$ and $W_n = \Theta(n)$, and instead just place weak restrictions on the allowed combinations of $W_n, H_n$ (e.g., the analysis of WBS in Fryzlewicz (2014) only requires $W_n H_n^2 \geq \log n$). These analyses (and those for all previously described estimators) are more refined than that given in Corollary 4: they are based on specific properties of the estimator in question. The corollary, on the other hand, follows from Theorem A.2, which uses a completely generic analysis that only assumes knowledge of the estimation error rate.*

**Remark A.3** (**Recovery in the Poisson model**). *Corollary 5 gives an approximate screening result for the post-processed fused lasso in the Poisson model, similar to the result in the strong sparsity, sub-Gaussian error case discussed above. As with all of our other approximate recovery results, this is established via the estimation error guarantees for the Poisson fused lasso estimator. Analyzing changepoint detection properties directly in the Poisson model seems like it could be a challenging task, and we are not aware of many results in the literature that do so. (Likewise for the binomial model; we did not state formal recovery results for this model but they follow from the estimation error bounds exactly as in the Poisson case, and changepoint detection analysis in this model seems difficult and we are not aware of extensive literature in this setting.)*

## A.11 Choosing a threshold level in the post-processing procedure

We describe a data-driven procedure to determine the threshold level $\tau_n$ of the filter in (24), used to derive a post-processed set of changepoints $S_F(\widetilde{\theta})$ from an estimate $\widetilde{\theta}$, as described in Theorem 5.

Let $\mathcal{A}(\cdot)$ denote a fitting algorithm that, applied to data $y$, outputs an estimate $\widetilde{\theta}$ of $\theta_0$ (e.g., $\mathcal{A}(y)$ could be the minimizer in (3), so that its output is the fused lasso estimate). In Algorithm 1 below, we present a heuristic but intuitive method for choosing the threshold level $\tau_n$, based on (entrywise) permutations of the residual vector $y - \widetilde{\theta}$. Aside from the choice of fitting algorithm $\mathcal{A}(\cdot)$, we must specify a number of permutations $B$ to be explored, the bandwidth $b_n$ for the filter in (24), and a quantile level $q \in (0, 1)$. The intuition behind Algorithm 1 is to set $\tau_n$ large enough to suppress "false positive" changepoints $100 \cdot q\%$ of the time (according to the permutations). This is revisited later, in the discussion of the simulation results.

Some example settings: we may take $\mathcal{A}(\cdot)$ to be the fused lasso estimator, where the tuning parameter $\lambda$ is selected to minimize 5-fold cross-validation (CV) error, $B = 100$, and $q = 0.95$. The choice of bandwidth $b_n$ is more subtle, and unfortunately, there is no one choice that works for all problems.[1] But, the theory in the last section provides some general guidance: e.g., for problems in which we believe there are a small number of changepoints (i.e., $s_0 = O(1)$) of reasonably large magnitude (i.e., $H_n = \Omega(1)$), Theorem 5 instructs us to choose a bandwidth that grows faster than $\log n(\log \log n)$, so, choosing $b_n$ to scale as $\log^2 n$ would suffice. We will use this scaling, and the above suggestions for $\mathcal{A}(\cdot)$, $B$, and $q$ in all coming experiments.

---

**Algorithm 1** Permutation-based approach for choosing $\tau_n$

---

0. Input a fitting algorithm $\mathcal{A}(\cdot)$, number of permutations $B$, bandwidth $b_n$, and quantile level $q \in (0, 1)$.
1. Compute $\widetilde{\theta} = \mathcal{A}(y)$. Let $\widetilde{S} = S(\widetilde{\theta})$ denote the changepoints, and $r = y - \widetilde{\theta}$ the residuals.
2. For each $b = 1, \ldots, B$, repeat the following steps:
   (a) Let $r^{(b)}$ be a random permutation of $r$, and define auxiliary data $y^{(b)} = \widetilde{\theta} + r^{(b)}$.
   (b) Rerun the fitting algorithm on the auxiliary data to yield $\widetilde{\theta}^{(b)} = \mathcal{A}(y^{(b)})$.
   (c) Apply the filter in (24) to $\widetilde{\theta}^{(b)}$ (with the specified bandwidth $b_n$), and record the largest magnitude $\widehat{\tau}^{(b)}$ of the filter values at locations greater than $b_n$ away from $\widetilde{S}$. Formally,
   $$\widehat{\tau}^{(b)} = \max_{\substack{i \in \{b_n, \ldots, n - b_n\}: \\ d(\widetilde{S}|\{i\}) > b_n}} \left| F_i(\widetilde{\theta}^{(b)}) \right|.$$
3. Output $\widehat{\tau}_n$, the level $q$ quantile of the collection $\widehat{\tau}^{(b)}$, $b = 1, \ldots, B$.

---

After running Algorithm 1 to compute $\widehat{\tau}_n$, the idea is to proceed with the filter $S_F(\widetilde{\theta})$, applied at the level $\tau_n = \widehat{\tau}_n$, to the estimate $\widetilde{\theta}$ computed on the original data $y$ at hand.

## A.12 Numerical simulations to verify some of our theoretical results

The code for the the results in this section can be found at `https://github.com/linnylin92/fused_lasso`. In our experiments, we use the following simulation setup. For a given $n$, the mean parameter $\theta_0 \in \mathbb{R}^n$ is defined to have $s_0 = 5$ equally-sized segments, with levels 0, 2, 4, 1, 4, from left to right. Data $y \in \mathbb{R}^n$ is generated around $\theta_0$ using i.i.d. $N(0, 4)$ noise. Lastly, the sample size $n$ is varied between 100 and 10,000, equally-spaced on a log scale. Figure A.1 displays example data sets with $n = 774$ and $n = 10,000$.

Figure A.1: *An example from our simulation setup for $n = 774$ (left) and $n = 10,000$ (right), where in each panel, the mean $\theta_0$ is plotted in red, and the data points in gray.*

For each sample size in consideration, we generated 50 example data sets from the setup described above, and on each data set, computed the full solution path of the fused lasso using the R package

genlasso. We applied 5-fold CV to determine $\lambda$, as implemented in genlasso: each consecutive, non-overlapping block of 5 points were grouped into 5 different folds. When minimizing the out-of-sample test mean squared error, the average of the immediate-left and immediate-right estimates were used as a proxy for the estimate at a particular location.

**Estimation error rate for fused lasso.** Figure A.2 displays the selected value of $\lambda$, as well as the estimation error $\|\widehat{\theta} - \theta_0\|_n^2$, averaged over the 50 trials, as functions of $n$. The results support the theoretical conclusion in Theorem 1, as the achieved estimation error rate scales as $\log n/n$ (perhaps even as $\log n(\log\log n)/n$, although it would be hard to tell the difference between the two). Also, CV appears to produce a choice of $\lambda$ that scales as $\sqrt{n}$, agreeing with the scaling of $\lambda$ prescribed by the theory. The screening distance $d(S(\widehat{\theta}) \,|\, S_0)$ was at most 5 across the entire simulation, regardless of $n$.

Figure A.2: *The left panel shows the median values of $\lambda$ chosen to minimize 5-fold CV error, aggregated over repetitions in our simulation setup, as the sample size $n$ varies. This scales approximately as $\sqrt{n}$, which is drawn as a red curve (with a best-fitting constant), supporting the choice prescribed by Theorem 1. The middle panel shows the corresponding estimation error $\|\widehat{\theta} - \theta_0\|_n^2$, again aggregated over repetitions, as $n$ varies. The scaling appears to be about $\log /n$ (red curve). The right panel plots the median values of $n\|\widehat{\theta} - \theta_0\|_n^2$ against $\log n$; this looks close to linear (red line), which provides empirical support to the claim that the fused lasso error rate is $\log n/n$ (or perhaps even $\log n(\log\log n)/n$, it would be hard to distinguish these two), which is roughly in agreement with Theorem 1. In each panel, vertical bars denote $\pm 1$ standard deviations.*

**Evaluation of the filter.** We demonstrate that the filter in (24), with $b_n = \lfloor 0.25\log^2 n \rfloor$, can be effective at reducing the Hausdorff distance between estimated and true changepoint sets. We first illustrate the use of the filter in a single data example with $n = 774$, in Figure A.3. As we can see, the fused lasso originally places a spurious jump around location 250, but this jump is eliminated when we apply the filter, provided that we set the threshold to be (say) $\tau_n = 0.5$.

Figure A.4 now reports the results from applying the filter in problems of sizes between $n = 100$ and $n = 10,000$, using 50 trials for each $n$. We consider three different sets of changepoint estimates: $S(\widehat{\theta})$, the original changepoints from fused lasso estimate $\widehat{\theta}$, tuning $\lambda$ via 5-fold CV; $S_F(\widehat{\theta})$, the changepoints after applying the reduced filter as described in Theorem 5 to $\widehat{\theta}$, with $\tau_n$ chosen by Algorithm 1; and $S_O(\widehat{\theta})$, an oracle set of changepoints given by trying a wide range of $\tau_n$ values and choosing the value that minimizes the Hausdorff distance after filtering (this assumes knowledge of $S_0$, and is infeasible in practice). These are labeled as "original", "data-driven", and "oracle" in the figure, respectively. As we can see from the left and middle panels, the Hausdorff distance achieved by the original changepoint set grows nearly linearly with $n$, but after applying the filter, the Hausdorff distance becomes very small, provided that $n$ is larger than 1000 or so. Empirically, the Hausdorff distance associated with the filtered set appears to grow very slowly with $n$, nearly constant (slower than the the $\log n(\log\log n)$ rate guaranteed by Theorem 5). The right panel shows that our data-driven choices of $\tau_n$ are not substantially different from those made by the oracle.

## Footnotes

[1]We note that in some situations, problem-specific intuition can yield a reasonable choice of bandwidth $b_n$. Also, it should be possible to extend Algorithm 1 to choose both $\tau_n$ and $b_n$, but we do not pursue this, for simplicity.