[Reviews · NeurIPS 2017]

Reviewer 1



The paper considers the 1d fused lasso estimator under squared loss. A new proof technique is introduced that yields tighter rates than prior work, and makes it possible to study extensions to misspecified and exponential family models. The paper is a pleasure to read, with a rich discussion of prior work and comparison against previous results. In view of Remark 3.7. however the rates obtained in the present work are not as sharp as those in Guntuboyina et al (2017) for the mean model. It would therefore be more transparent, if the authors mention this upfront, e.g. in section 2. In the present version of the manuscript, Remark 3.7 comes about as a surprise and one feels disappointed... The strength of the contributions here is the idea of the lower interpolent. This should be better emphasized, e.g. by providing a gist of this concept e.g. line 81 rather than leaving the definition buried in the proof sketch. If possible, it would be important to provide some explanation as to why the authors think that the proof technique of Guntuboyina et al (2017) cannot be extended to misspecified and exponential family models. As future work it would be interesting to see whether the lower interpolent technique can be used to analyze the fused lasso estimator in the context of linear regression

Reviewer 2



The authors study the problem of multiple change point detection and provide provide bounds for the fused-lasso estimator. The problem of change point detection is well studied in the statistics literature and the authors provide good references. The paper is well written and flows well. As claimed by the authors, one of the major contribution of this paper is in the technique of the proof and, unfortunately, the authors had to put it in the appendix to fit the NIPS page limits. I can not think of a way to fit that in the main section without a major rewrite. In its current form, the appendix is longer than the main paper. A recent independent development, Guntuboyina et al. (2017), provide the bounds for the lasso estimator. The authors claim that their method of proof is of independent interest as that generalizes to other related problems. It is not clear why the methods of this generalizes but that of Guntuboyina et al. (2017) does not. I would have loved to see discussions on how theorem 3.1 compares to the main results in Guntuboyina et al. (2017) which seem to be stronger. A comment on inequality (7): This seems to follow directly from the definition of \hat\theta unless I am missing something. Isn’t \hat\theta defined to be the parameter that minimizes 1/2 ||y-\theta||_2^2 + \lambda ||D\theta||_1

Reviewer 3



This paper gives a sharp error bound for the fused lasso estimator. To give a sharper bound, a new concept "lower interpolant" is introduced. The derived bound improves the existing bound and is applicable also to a misspecified model and an exponential family. I feel that the paper gives an interesting theoretical tool. The given bound gives significantly improved rate for especially small W_n (the shortest interval between the break points). One drawback is that the generality of the lower interpolant is not clear. If it could be applied for other types of structured regularizations, then the result would be more significant.